# Fuzzy supertertiary interactions within PSD-95 enable ligand binding

George L Hamilton[1†‡], Nabanita Saikia[1†§], Sujit Basak[2], Franceine S Welcome[2], Fang Wu[2], Jakub Kubiak[3], Changcheng Zhang[2], Yan Hao[2], Claus AM Seidel[3], Feng Ding[1]*, Hugo Sanabria[1]*, Mark E Bowen[2]*

[1]Department of Physics and Astronomy, Clemson University, Clemson, United States; [2]Department of Physiology and Biophysics, Stony Brook University, Stony Brook, United States; [3]Molecular Physical Chemistry, Heinrich Heine University, Düsseldorf, Germany

**Abstract** The scaffold protein PSD-95 links postsynaptic receptors to sites of presynaptic neurotransmitter release. Flexible linkers between folded domains in PSD-95 enable a dynamic supertertiary structure. Interdomain interactions within the PSG supramodule, formed by PDZ3, SH3, and Guanylate Kinase domains, regulate PSD-95 activity. Here we combined discrete molecular dynamics and single molecule Förster resonance energy transfer (FRET) to characterize the PSG supramodule, with time resolution spanning picoseconds to seconds. We used a FRET network to measure distances in full-length PSD-95 and model the conformational ensemble. We found that PDZ3 samples two conformational basins, which we confirmed with disulfide mapping. To understand effects on activity, we measured binding of the synaptic adhesion protein neuroligin. We found that PSD-95 bound neuroligin well at physiological pH while truncated PDZ3 bound poorly. Our hybrid structural models reveal how the supertertiary context of PDZ3 enables recognition of this critical synaptic ligand.

**\*For correspondence:**
fding@clemson.edu (FD);
hsanabr@clemson.edu (HS);
mark.bowen@stonybrook.edu (MEB)

†These authors contributed equally to this work

**Present address:** ‡Department of Biochemistry and Molecular Pharmacology, New York University School of Medicine, New York, United States; §Chemistry Department, New Mexico Highlands University, Las Vegas, United States

**Competing interest:** The authors declare that no competing interests exist.

## Editor's evaluation

This paper presents data of fundamental importance and is methodologically exceptional. It is of broad interest to investigators studying the function and regulation of protein scaffolds, dynamic protein structure, and the regulation of the postsynaptic density at excitatory synapses. The authors develop an integrated approach using fluorescence-based biochemical methods, disulfide mapping, and discrete molecular dynamic simulations to study the dynamic supertertiary conformation of the synaptic scaffold protein PSD-95. The overall research strategy serves as a textbook example to the field.

## Introduction

The fundamental structural unit of large proteins is the independently folding domain (*Han et al., 2007*). Most proteins are composed of more than one domain (*Ekman et al., 2005*; *Apic et al., 2001*). Evolution has shuffled the deck to produce a rich diversity of multidomain proteins. A prototypical example is the Membrane Associated Guanylate Kinases (MAGuKs), which contain an array of folded protein-interaction domains connected in series. MAGuKs are scaffold proteins that link cell surface membrane proteins to their intracellular signaling partners and the cytoskeleton (*Ye et al., 2018*). MAGuK family members control diverse processes ranging from epithelial cell polarity (*Funke et al., 2005*) to synaptic neurotransmitter signaling (*Won et al., 2017*).

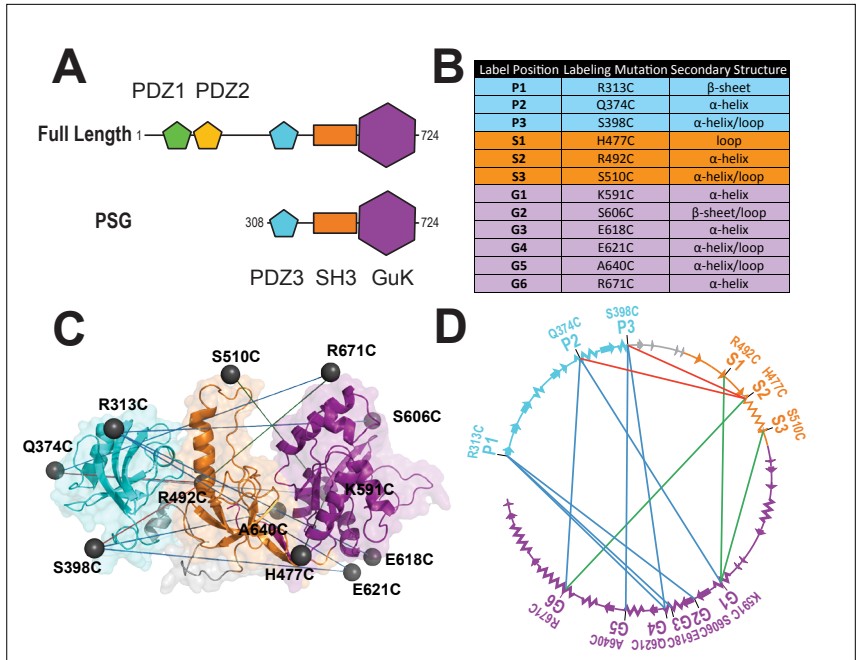

**Figure 1.** A FRET network to probe structure and dynamics in PSD-95. (**A**) Schematic representation of the domain organization in PSD-95. (**B**) Cysteine substitutions used for fluorescent labeling. Label position indicates the domain for each mutant and the order in the primary sequence. Label mutation gives the residue identity. Secondary structure describes the local secondary structural environment at each site. (**C**) Location of the labeling sites within the supertertiary structure of the PSG fragment. The PSG is shown in cartoon representation with PDZ3 (cyan), SH3 (orange) and GuK (purple). Spheres indicate the location of the dye while lines indicate the experimental Förster resonance energy transfer (FRET) pairs. Although only the PSG is shown, measurements were made on full-length PSD-95. (**D**) Connectivity in the FRET network. The primary sequence of PSD-95 is shown in a circular representation with each domain colored as in panel C. The secondary structural elements are indicated: α helices (zig zag) and β sheets (arrows). The position of each labeling site is indicated by the domain and order in the primary sequence. The FRET pairs used for measurements are indicated by lines connecting the labeling sites used in that variant with FRET pairs spanning PDZ3-GuK (blue), PDZ3-SH3 (red), and SH3-GuK (green).

Almost all MAGuKs contain a conserved 'PSG' supermodule that links a Postynaptic Density Protein of 95 kDa, Discs Large, Zonula Occludens-1 (PDZ) domain (*Ye and Zhang, 2013*) an Src Homology 3 (SH3) domain (*Saksela and Permi, 2012*) and Guanylate kinase-like (GuK) domain (*Kistner et al., 1995*; *Olsen and Bredt, 2003*) that serves as another protein-binding domain (*Kim et al., 1997*; *Figure 1A*). One distinctive feature of the MAGuKs is the insertion of a variable HOOK (*Hough et al., 1997*) region within the SH3 domain that disrupts canonical SH3 domain interactions (*McGee et al., 2001*). Different MAGuKs append additional protein-binding domains to this PSG supermodule (*Funke et al., 2005*). The postsynaptic density protein of 95 kilodaltons (PSD-95) contains an N-terminal extension and two tandem PDZ domains attached to its PSG (*Figure 1A*).

In multidomain proteins, the proximity imparted by interdomain linkers results in high effective concentrations of the folded domains (*Sorensen and Kjaergaard, 2019*). This can result in specific domain interactions that are too weak to manifest when the domains are not connected (*McCann et al., 2011*). Linkers allow for different supertertiary arrangements but also limit the conformational landscape permitting only a subset of supertertiary structures to exist (*Tompa, 2012*). When proteins adopt multiple conformations, they must be represented by an ensemble of states. The presence of heterogeneity confounds structural biology methods reliant on ensemble averaging, while protein dynamics leads to time averaging even in single-molecule experiments (*Medina et al., 2021*).

Mapping the conformational landscape of the PSG is important because these supertertiary interactions regulate protein interactions with scaffolding clients (*Rademacher et al., 2019*; *Rademacher et al., 2013*; *Qian and Prehoda, 2006*). Previous SAXS and NMR studies along with computational approaches have proposed binding sites for PDZ3 within SH3-GuK (or a lack thereof) (*Zhang et al.,*

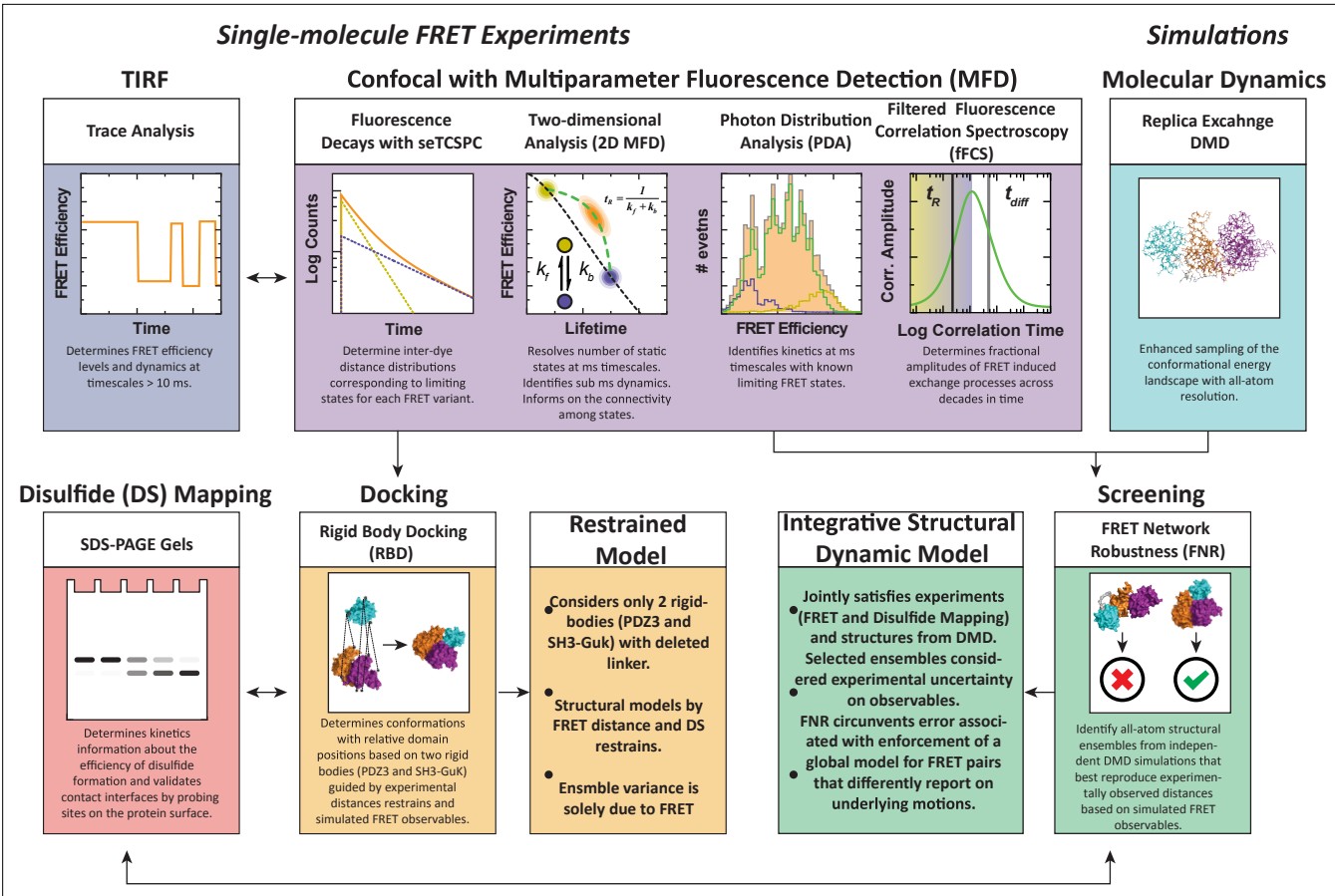

**Figure 2.** An experimental workflow for dynamic structural biology. For single molecule Förster resonance energy transfer (smFRET) experiments, total internal reflection fluorescence (TIRF) and confocal modalities were used to probe the FRET network consisting of 12 complementary pairs of FRET labeling sites. Comparing the two approaches provides validation through the use of different dyes, sample preparation, instrumentation, and handling. Confocal smFRET with pulsed interleaved excitation and multiparameter fluorescence detection were holistically analyzed in four modalities, including fluorescence decay analysis, two-dimensional representations, photon distribution analysis, and filtered fluorescence correlation spectroscopy. Each approach provides complementary information on the underlying conformational distribution with respect to inter-dye distances and protein dynamics. For simulations: replica-exchange discrete molecular dynamics (rxDMD) provided an independent view of the energy landscape. For structural modeling: two approaches were used. Rigid body docking used two fixed structures (PDZ3 and SH3-GuK) to identify domain orientations that satisfied the FRET-derived interdye distances for all 12 FRET pairs, resulting in a fully FRET-restrained model. Screening of the rxDMD simulations with the FRET distances and the boundaries from FRET robustness analysis allowed classification of structures that mutually satisfy the simulated model (rxDMD) and the FRET observables. Finally, to validate the proposed structural models, we used disulfide mapping and probed interfaces corresponding to the preferred structural basins. In a refinement step, we introduced the disulfide information for both structural modeling approaches.

*2013*; *Korkin et al., 2006*; *McCann et al., 2012*). Single-molecule Förster resonance energy transfer (smFRET) studies revealed that PDZ3 was dynamic but had a defined orientation relative to SH3-GuK (*McCann et al., 2012*).

We have developed a workflow for dynamic structural biology that combined single molecule fluorescence measurements with simulations and disulfide mapping (*Figure 2*), which we recently used to characterize the PDZ tandem from PSD-95 (*Yanez Orozco et al., 2018*). In brief, single molecule confocal microscopy data is collected using multiparameter fluorescence detection (MFD) and analyzed in several ways to extract structural and dynamic details used in modeling. The single molecules containing active donor and acceptor are selected for sub-ensemble time-correlated single photon counting (seTCSPC) to generate fluorescence intensity decays. The intensity decays are fit to assign the number of conformational states along with distances between the dyes in each state. Total internal reflectance fluorescence microscopy (smTIRF) is used to benchmark the time-averaged FRET distributions using alternate FRET dyes and probe for slow timescale dynamics.

To generate structural models, the FRET distances from fitting the donor fluorescence decays, along with accessible volume (AV) simulations of the dyes at each labeling site, are used as restraints in rigid body docking. This can position the known structures of individual domains to generate supertertiary structural models of each limiting state. To assign dynamic timescales for exchange between states, the confocal microscopy data is further analyzed using filtered fluorescence correlation spectroscopy (fFCS). Additionally, photon distribution analysis (PDA) is used to obtain relaxation rate constants for conformational exchange on timescales close to the diffusion time. To provide an independent picture of the conformational energy landscape, we performed replica exchange discrete molecular dynamics (DMD) simulations. We compare structural models from DMD to FRET models from rigid body docking and use the DMD models to examine the unique sensitivity of each FRET pair to the underlying conformational ensemble.

Here, we applied this approach to the PSG supramodule from PSD-95. Experiments and simulations agreed and revealed two distinct conformational basins for PDZ3; a fuzzy interaction with PDZ3 within a broad interface near the SH3 HOOK insertion and a second discrete binding site in GuK. Both were confirmed with disulfide mapping. Surprisingly, these supertertiary interactions in full-length PSD-95 allowed PDZ3 to interact with the synaptic adhesion protein neuroligin, a known binding partner of PDZ3 (*Irie et al., 1997*; *Fu et al., 2003*; *Saro et al., 2007*), while the truncated domain showed weaker binding. Thus, the supertertiary context enhanced the binding activity of PDZ3 towards a critical physiological ligand.

Our integrative approach to dynamic structural biology resolved the structural heterogeneity of the PSG supramodule within full-length PSD-95. Combining simulation with experiments resolved global dynamics and provided residue-level details about supertertiary interactions. This approach moves beyond solving a single structure towards resolving an ensemble of conformers with differing behavior and is applicable to other dynamic multidomain proteins.

## Results

### Mapping the supertertiary conformational landscape with single-molecule FRET

To experimentally probe the location of PDZ3 within the PSG supramodule, we used 12 cysteine mutations spanning PDZ3, SH3, and GuK (*Figure 1B*). These labeling sites were identified in our previous work, which included screening of dye properties at these sites through measurements of ensemble anisotropy and quantum yield (*McCann et al., 2012*). The labeling sites that we published have been used alone or in various combinations to produce more than one hundred labeling variants. Each of the labeling variants was further screened to identify any changes in protein expression or purification that differ from the wild type PSD-95, which would be indicative of misfolding. All of the variants used in this work were homogeneous with respect to their chromatographic properties and monodisperse as assessed by our size exclusion chromatography before measurements.

By using these labeling sites in different combinations, we created 12 variants (*Figure 1C*). These variants form a FRET network, which is necessary for structural modeling (*Figure 1D*; *Medina et al., 2021*). We used two experimental approaches to measure smFRET. The same batches of purified PSD-95 variants were split for measurements in solution using confocal microscopy with multiparameter fluorescence detection (MFD) and also encapsulated within lipid vesicles for measurements using smTIRF. These techniques use different labeling reactions with different dye pairs to ensure that the results are not affected by protein handling or experimental modality.

Fluorescently labeled proteins measured with confocal microscopy with pulsed-interleaved excitation (PIE) (*Kudryavtsev et al., 2012*), which allowed selection of molecules with active donor and acceptor fluorophores (*Appendix 1—table 1* and *Appendix 1—table 2*). This selected population was used for subensemble time-correlated single photon counting (seTCSPC) to generate the fluorescence intensity decays for the FRET samples and their matching donor-only controls (*Figure 3A*; *Dimura et al., 2016*). One of the necessary assumptions in this work is that there is one conformational landscape for PSD-95 (i.e. ground truth) and that each variant provides specific information on the same landscape. This includes the assumption, based on photophysical and chromatographic measurements, that the dyes do not significantly perturb the protein conformation. To capture this shared information, we performed a simultaneous global analysis of all variants (*Opanasyuk et al.,*

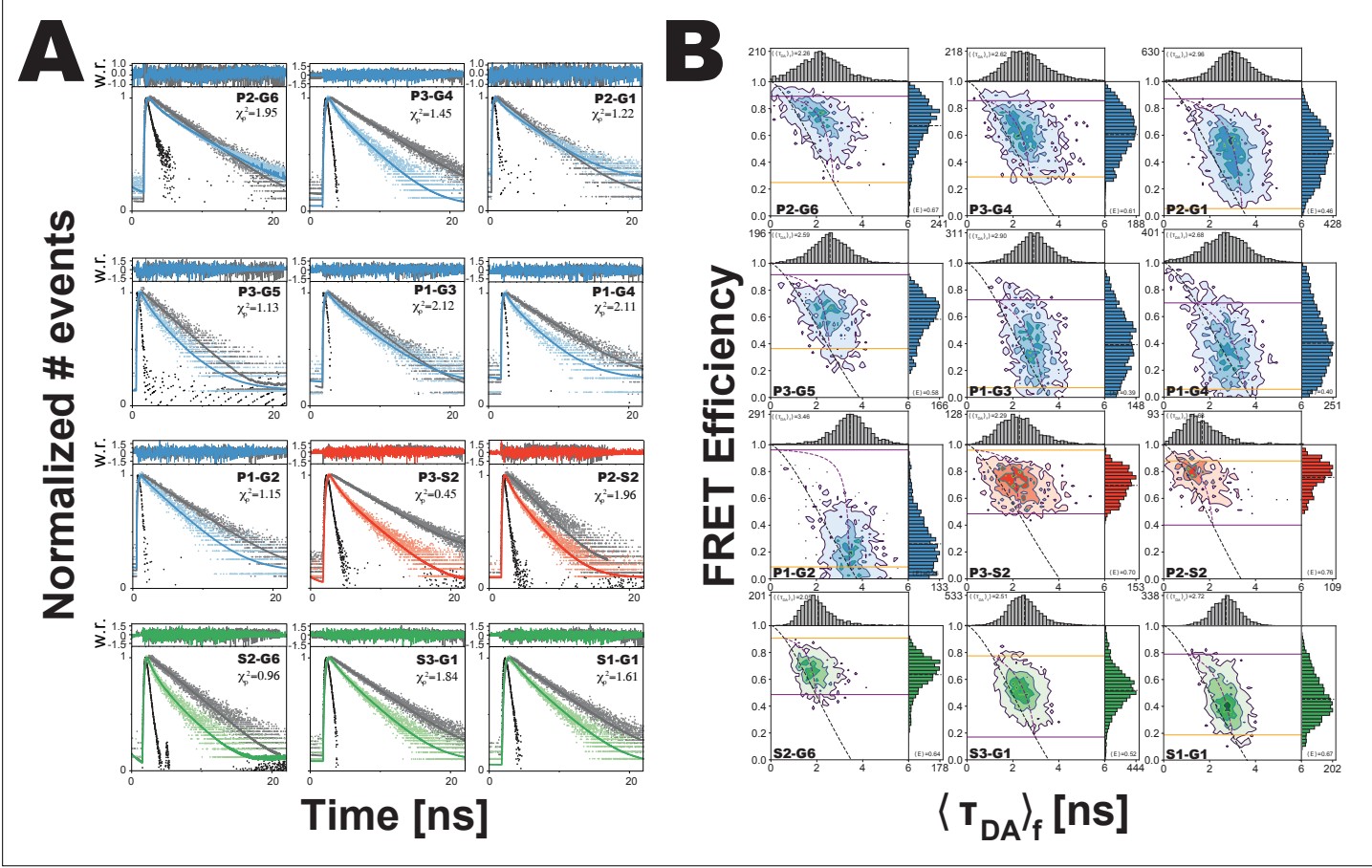

**Figure 3.** Analysis of confocal microscopy data. (**A**) Global fit of the Förster resonance energy transfer (FRET)-sensitized donor fluorescence decay curves for full length variants. Shown are subensemble time correlated single photon counting decays for FRET pairs spanning PDZ3-GuK (blue), PDZ3-SH3 (red), and SH3-GuK (green). The instrument response function is shown in black, and the donor-only fluorescence decay is shown in gray. Raw histogram data are shown as points, with fits overlaid as lines. FRET quenches donor fluorescence, which reduces the lifetime and introduces curvature into fluorescence decay. The presence of more than one underlying state with different FRET efficiencies results in multi-exponential fluorescence decays. Fit parameters can be found in *Appendix 1—table 2* and *Appendix 1—table 3*. Details of the model and fit can be found in Materials and methods. (**B**) Multiparameter fluorescence histograms of full-length PSD-95 FRET variants. Multiparameter fluorescence detection histograms for FRET pairs spanning PDZ3-GuK (blue), PDZ3-SH3 (**red**), and SH3-GuK (green) as noted in each panel. Variant details can be found in *Figure 1*. Overlaid on the contour plots are the static FRET-lines (black, dashed), dynamic FRET-lines (purple, dashed), and solid horizontal lines corresponding to the limiting states A (orange) and B (purple) from seTCSPC. Also given are the burst-wise average values for the mean donor fluorescence lifetime ($\langle\langle \tau_{DA}\rangle_f\rangle$) and mean FRET efficiency ($\langle E \rangle$) (black, dashed lines in 2D histograms). Dynamic exchange is immediately evident from broadening and skew rightward from the static FRET-line for each variant. Correction parameters for these histograms can be found in *Appendix 1—table 2* while parameters for the static and dynamic FRET-lines can be found in *Appendix 1—table 5*, *Appendix 1—table 6*, respectively.

*2022*). The FRET distances were variant-specific while the number and occupancy of conformational states were set as global fitting parameters. This global constraint is how we implement our assumption that the conformational distribution is the same for all variants, but that each variant senses the underlying conformations differently.

To determine the number of conformational states present in the ensemble, we compared global fits using models that included a no-FRET state along with an increasing number of FRET states (*Table 1*). We observed a significant improvement in the fitting statistics for all variants when we increased from one FRET state to two. Adding a third global FRET state marginally improved the individual fit to three variants while the remaining nine variants remain unchanged or showed a slightly worse fit. Thus, based on fitting statistics (*Table 1*), we conclude that a two-state model with a small no FRET population is sufficient to fit all the data from all 12 variants (*Appendix 1—table 3*). This two state model suggests that PDZ3 samples two limiting conformations with slight predominance of state

**Table 1.** Fit statistics for models with increasing numbers of states.

The fit parameter $\chi^2_{r,seTCSPC}$ is shown for each variant when fit to a model with an increasing number of structural states as indicated above each column. Increasing from one to two states results in a significant increase in fit quality. Further increase to three states provides little to no improvement in fit quality. Increasing model complexity beyond two states decreases the average fit quality. Details of variants can be found in *Figure 1*.

| Sample | One State $\chi^2_{r,seTCSPC}$ | Two State $\chi^2_{r,seTCSPC}$ | Three State $\chi^2_{r,seTCSPC}$ |
|---|---|---|---|
| P2-G6 | 8.46 | 2.54 | 1.95 |
| P3-G4 | 3.32 | 1.49 | 1.67 |
| P2-G1 | 1.66 | 1.22 | 1.53 |
| P3-G5 | 1.55 | 1.13 | 1.45 |
| P1-G3 | 2.86 | 2.12 | 2.38 |
| P1-G4 | 2.87 | 2.11 | 2.31 |
| P1-G2 | 1.54 | 1.15 | 1.52 |
| P3-S2 | 0.68 | 0.45 | 0.65 |
| P2-S2 | 14.65 | 1.96 | 1.58 |
| S2-G6 | 1.77 | 0.92 | 1.23 |
| S3-G1 | 5.75 | 1.84 | 1.78 |
| S1-G1 | 2.94 | 1.61 | 1.71 |
| Average | 4.00 | 1.55 | 1.65 |

B (53.9%) over state A (46.1%). The global fit assigns all 24 distances to their respective states, which solves the challenge of uniquely assigning both states for each variant (*Appendix 1—table 4*). We note that these fits all recovered a single diffusion time indicating a lack of higher order multimers, which is consistent with our previous analytical size exclusion chromatography showing that these constructs are monodisperse (*McCann et al., 2012*).

For each variant, we plotted the intensity-based FRET efficiency against the average donor fluorescence lifetime ($\langle\tau_{D(A)}\rangle_f$) for each molecule (*Figure 3B*). We calculated the expected relationship between FRET intensity and lifetime using the assumption of no dynamics, which we plotted as the static FRET-line (*Barth et al., 2022*; *Appendix 1—table 5*). Molecules undergoing dynamics between states with different lifetimes would fall to the right of this line. The dynamic FRET-lines represent all possible degrees of mixing between the limiting states (*Appendix 1—table 6*). All measured variants exhibited a rightward skew away from the static FRET-line, a hallmark of conformational dynamics (*Figure 3B*; *Barth et al., 2022*). The variants are positioned differently along the dynamic FRET-line, which is a function of the underlying rate constants for the structural transition along with the unique sensitivity of each variant to that transition.

Variants involving PDZ3 exhibited broad or irregular distributions indicating a heterogeneous conformational ensemble. Variants between PDZ3 and SH3 exhibited higher FRET efficiency, suggesting close proximity. The SH3-GuK domains have restricted interdomain motion (*McGee et al., 2001*; *Tavares et al., 2001*). Nonetheless, FRET variants spanning SH3-GuK still fell off the static FRET-line. These variants exhibited narrower, more regular FRET distributions relative to PDZ3-labeled variants indicating fast but limited dynamics within SH3-GuK.

## Comparison of full-length PSD-95 to the PSG truncation

To probe whether interactions within PSD-95 affect the PSG, we also measured 6 of the FRET variants in a truncated PSG fragment. Measurements using smTIRF with camera detection revealed changes in the time-averaged FRET distributions for all variants (*Figure 4A* and *Figure 4—figure supplement 1*). Truncating PSD-95 resulted in broader and more multi-peaked distributions. For example, variant P1-G3 (*Figure 1B*), splits into lower and higher FRET species in the truncation. Similarly, variant P3-S2

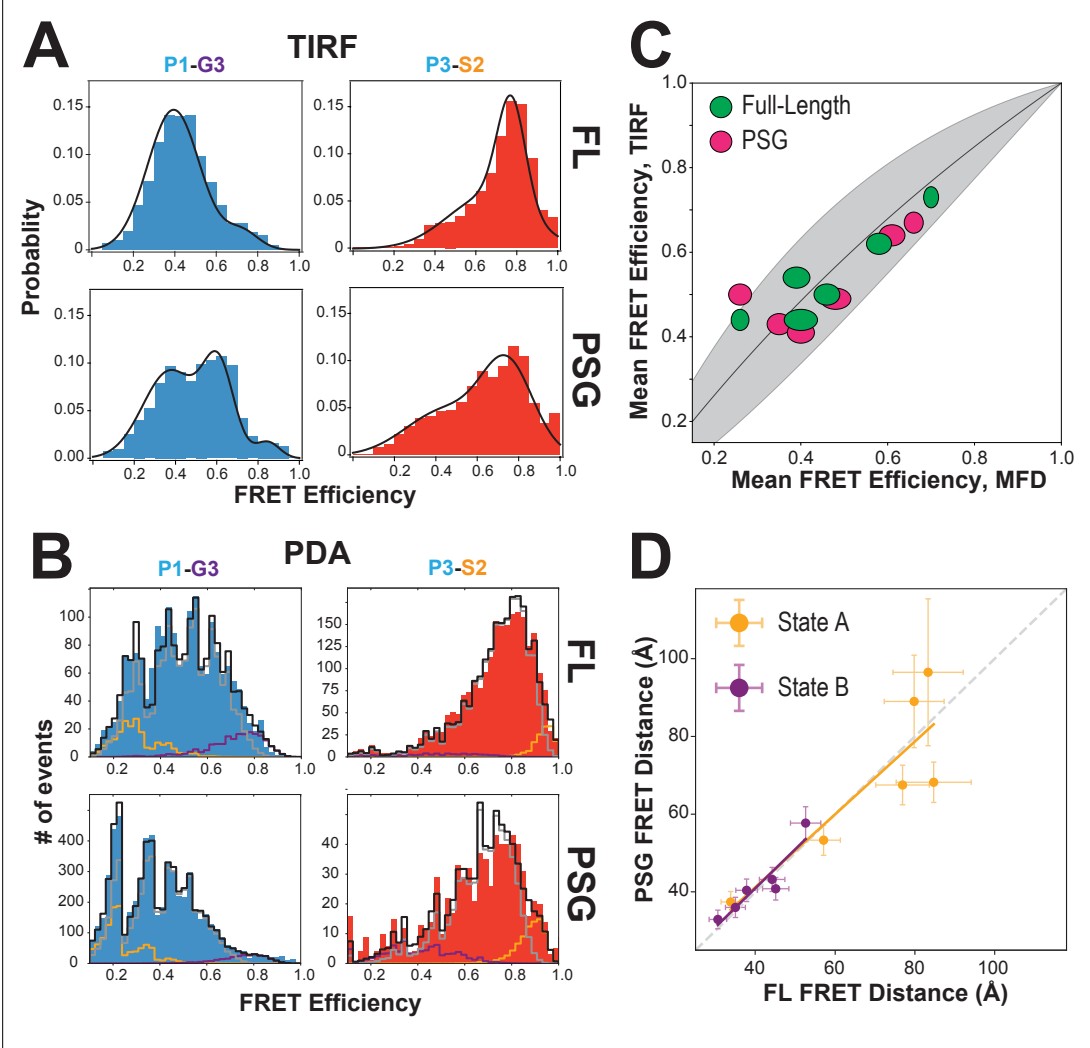

**Figure 4.** Effect of truncating PSD-95 on PSG supertertiary structure. (**A**) Representative single-molecule total internal reflection fluorescence (smTIRF) Förster resonance energy transfer (FRET) efficiency histograms for full-length PSD-95 (top) and the corresponding truncated PSG fragment (bottom). Shown are variants P1-G3 (blue) and P3-S2 (red). (**B**) Representative PDA plots using a 2 ms time window for the same variants from panel A using the same coloring. Molecules occupying limiting states A and B are highlighted in orange and purple, respectively. (**C**) Comparison of mean FRET efficiency as measured with smTIRF (y-axis) and multiparameter fluorescence detection (MFD) (x-axis) for full-length (green) and PSG (pink) variants. Ellipse eccentricities represent the relative width of FRET distributions observed by each method. The expected relationship given the different fluorophores used is shown as a line with the shaded region corresponding to Förster radius uncertainty. Förster radii used were as used previously (***McCann et al., 2012***; ***Yanez Orozco et al., 2018***). The fit to the ideal relationship gave $\chi^2_{ALL}$ = 2.13 with $\chi^2_{FL}$.=0.72 and $\chi^2_{PSG}$ = 1.42. (**D**) Comparison of the seTCSPC limiting-state distances for full-length PSD-95 (y-axis) and the PSG fragment (x-axis). Distances are shown for state A (orange; Slope = 0.94; Pearson Correlation Coefficient ($R_p$)=0.86) and state B (purple; Slope = 1.0; $R_p$ = 0.93).

The online version of this article includes the following source data and figure supplement(s) for figure 4:

**Source data 1.** This zip archive contains all of the FRET values for each variant measured with smTIRF.

**Figure supplement 1.** Effect of truncation on PSD-95 as measured with smTIRF.

**Figure supplement 2.** Comparison of multiparameter fluorescence histograms of PSG-truncated and full-length PSD-95 Förster resonance energy transfer (FRET) variants.

**Figure supplement 3.** Global fit of seTCSPC Förster resonance energy transfer (FRET)-sensitized donor fluorescence decay curves for truncated PSG variants.

**Figure supplement 4.** Photon distribution analysis histograms.

showed the highest FRET efficiency but the distribution spread out to lower FRET efficiency when PSD-95 is truncated. Truncation also increased anticorrelated FRET transitions in smTIRF time traces suggesting altered dynamics of PDZ3 (*Figure 4—figure supplement 1C*). To quantify this, we determined the Pearson correlation coefficients for donor-acceptor intensities for each molecule. While the magnitude of the Pearson correlation coefficient depends on FRET efficiency and exchange rate constants, comparison of the same labeling sites in full-length and PSG revealed a uniform increase in FRET transitions when PSD-95 is truncated (*Figure 4—figure supplement 1D*).

Confocal measurements also revealed shifts in FRET efficiency and donor lifetime for most variants (*Figure 4—figure supplement 2*, *Appendix 1—table 3*, *Appendix 1—table 4*). To further analyze the intensity-based FRET efficiencies from confocal measurements, we performed photon distribution analysis (PDA) (*Kalinin et al., 2007*; *Figure 4B*). Our fit model included two static, limiting states and a dynamic population (*Appendix 2—table 1*). Several truncated variants had more molecules in static limiting states, indicating increased dwell time of PDZ3 (*Appendix 2—table 1*). This suggests the slowest exchange processes were further slowed in truncated variants. However, the predominant state for PDZ3 was always in exchange with faster relaxation rates. The truncation-induced shifts in FRET efficiency were similar to smTIRF (*Figure 4*). The good agreement between mean FRET efficiencies measured with smTIRF and MFD, representing the long-time averages from both techniques, brings additional confidence in the results (*Figure 4C*). A global fit of fluorescence decays for the PSG recovered two states similar to the full-length protein (*Figure 4D*, *Appendix 1—table 3*, *Appendix 1—table 4*). Truncating PSD-95 shifted the limiting state distances for state A and slightly reduced state B occupancy (*Appendix 1—table 4*).

To resolve fast conformational dynamics, we performed filtered Fluctuation Correlation Spectroscopy (fFCS) (*Felekyan et al., 2012*). We filtered bursts into subensembles representing the limiting states and analyzed these components using correlation algorithms (*Felekyan et al., 2012*; *Figure 5—figure supplement 1*). Just as FRET efficiency differed between variants, each variant is differently attuned to the same underlying conformational transitions, so data was fit globally to capture the shared information (*Appendix 3—table 1*). Three decay times were assigned to local motions ($t_{R1}$) that maintain residue contacts, domain reorientations ($t_{R2}$) that alter interdomain interaction interfaces, and domain exchange ($t_{R3}$) such as large-scale translational exchange between basins (*Figure 5A*). PSD-95 variants displayed complex dynamics with components from microseconds to milliseconds. To highlight differences between variants, we plotted the normalized relaxation amplitudes in a matrix representation, which is the dynamics equivalent of a contact map for protein interactions (*Figure 5B & D*). This revealed that 7 out of the 10 GuK-labeled variants have major (red) or middle (yellow) contributions at $t_{R1}$. The remaining three variants are dominated by $t_{R3}$. The large contribution at $t_{R1}$ for most PDZ3-GuK variants suggests fast local motions within the limiting states, while for variants P2-G5 and P3-G5 domain exchange dominated the dynamics. We also note that for four of the remaining 5 PDZ3-GuK variants, the $t_{R3}$ has middle or major contributions. Moreover, variants P2-G5, P3-G4, and P3-G5 reported middle contributions from domain reorientation.

Summarizing the dynamics observed for the PDZ3-GuK variants, the major contributions are either $t_{R1}$ suggesting fast domain motions within identified basins (4 out of 7 variants) or $t_{R3}$ indicating slow jumps between conformational basins (3 out of 7 variants). Three variants had their middle contribution at $t_{R2}$ arising from domain reorientations. Truncated PSG variants exhibited increased heterogeneity in dynamics (*Figure 5C & E*) although the major or middle contributions to dynamics appear at $t_{R1}$ or $t_{R3}$.

## Discrete molecular dynamics Simulations of the PSG core

To map the conformational energy landscape of the truncated PSG supramodule, we performed replica exchange DMD simulations using 18 replicas running at neighboring temperatures with a cumulative total simulation time of 11.9 μs. To avoid bias, we chose an extended starting conformation with PDZ3 not in contact with SH3-GuK (*Figure 6—figure supplement 1*). The probability density function of the radius of gyration ($R_g$) shows that PDZ3 did not linger in extended conformations, which were rarely sampled (*Figure 6A* and *Figure 6—figure supplement 2*). Instead, the PDZ3 primarily adopted a docked medium conformation (α) with a mean $R_g$ of 27.6 Å along with a more compact conformation (β) with mean $R_g$ of 23.4 Å. Representative models from these three populations (extended, medium,

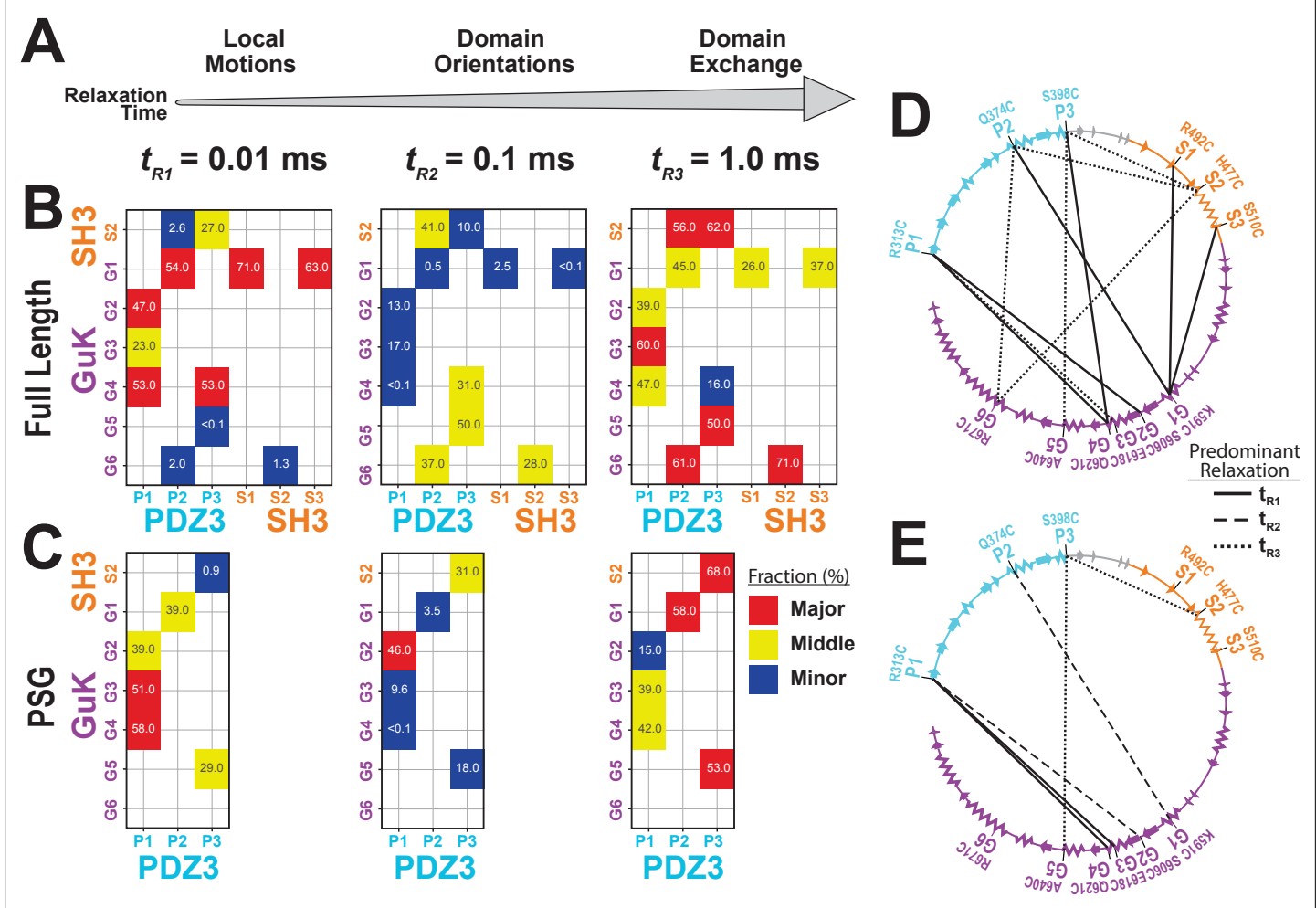

**Figure 5.** Effect of truncating PSD-95 on supertertiary dynamics within PSG. Site-specific dynamics for each variant as measured with filtered fluorescence correlation spectroscopy (fFCS) and fit to a model with three correlation times at three decades in time from $10^{-5}$ s to $10^{-3}$ s. (**A**) Association between the dynamic relaxation time and the expected protein motions across the three decades in time used in the analysis. (**B & C**) Matrix representations of the relative contribution to dynamic relaxation at each decade in time (as indicated above the panel). The axes specify the domains and labeling sites with each variant placed at the intersection between sites used. In both panels, the relative contribution to relaxation at each timescale is highlighted: major (red); middle (yellow); minor (blue). The percentage is shown within each square. (**B**) Extent of dynamics across timescales for full length PSD-95. (**C**) Extent of dynamics across timescales for the PSG truncation, measured with the same variants. (**D & E**) The major contribution to dynamics is mapped onto the primary sequence. Shown are circular representations of the primary sequence with PDZ3 (cyan), SH3 (orange) and GuK (purple). The secondary structural elements are indicated: α helices (zig zag) and β sheets (arrows). Lines denote the fluorophore labeling sites for each measurement and indicate the predominant relaxation timescale: $t_{R1}$ (solid); $t_{R2}$ (dashed); $t_{R3}$ (dotted). (**D**) Predominant timescale for relaxation of each FRET variant measured in full length PSD-95. (**E**) Predominant timescale for relaxation of each FRET variant measured in the PSG truncation. The most dominant relaxation times, $t_{R1}$ and $t_{R3}$, correspond to fast local motions and domain-scale conformational exchange, respectively.

The online version of this article includes the following figure supplement(s) for figure 5:

**Figure supplement 1.** Filtered fluorescence correlation spectroscopy fits.

and compact) reveal a diverse ensemble of conformations with PDZ3 sampling both SH3 and GuK as well as undocked conformations (***Figure 6B***).

To represent the cumulative association of PDZ3 with SH3 and GuK, we plotted the distance between centers of mass (COM) for PDZ3 and SH3 against the distance between COMs for PDZ3 and GuK (***Figure 6C***). This 2D free energy profile depicts a broad low-energy basin with PDZ3 closer to SH3 (***Figure 6C***, α-basin). This ensemble corresponds to the predominant population in the $R_g$ distribution. Within the α-basin, PDZ3 localized to the HOOK insertion rather than the canonical SH3 domain (***Figure 6D***). PDZ3 also samples a shallower basin closer to GuK, which is separated by an energy barrier of ~2.0 kcal/mol (***Figure 6C***, β-basin). This population corresponds to the compact

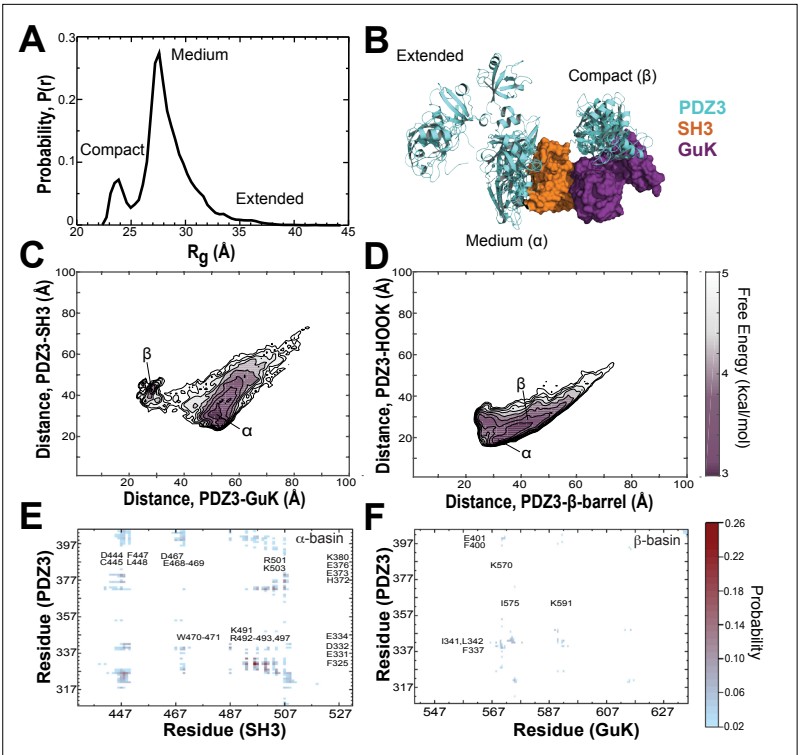

**Figure 6.** Discrete molecular dynamics of the PSG supramodule from PSD-95. (**A**) Probability distribution of the radius of gyration ($R_g$) of the PSG during replica-exchange discrete molecular dynamics (DMD) simulations totaling 11.9 μs. (**B**) Representative conformations from the three basins apparent in the $R_g$ distribution. The conformations and their respective population fractions in the highly sampled α-basin and less frequently sampled β-basin are provided in *Figure 6—figure supplement 3*. (**C**) 2D free energy landscape of the relative distance between centers of mass (COM) for PDZ3 and GuK (x-axis) or SH3 β-barrel (y-axis). (**D**) 2D free energy landscape of the relative distance between COM for PDZ3 and either the SH3 β-barrel (x-axis) or the SH3 HOOK insertion (y-axis). (**E**) Probability of pairwise residue contacts between PDZ3 and SH3, which define the α-basin. Residues from PDZ3 are on the y-axis while residues from SH3 are on the x-axis. (**F**) Probability of pairwise residue contacts between PDZ3 and GuK, which define the β-basin. Residues from PDZ3 are on the y-axis while residues from GuK are on the x-axis. The associated color bar indicates the normalized probability of the individual pairwise interactions.

The online version of this article includes the following figure supplement(s) for figure 6:

**Figure supplement 1.** The starting conformation for the PSG supramodule from PSD-95 used in DMD simulations.

**Figure supplement 2.** Time evolution of the radius of gyration ($R_g$) of PSG supramodule for 18 replicas discrete molecular dynamics (DMD) simulations.

**Figure supplement 3.** Representative conformations of PSG supramodule from PSD-95 (PDBDEV_00000164).

**Figure supplement 4.** Comparison of equilibrated conformations observed in discrete molecular dynamics (DMD) to published crystal structures.

configuration in the $R_g$ distribution. In addition to these two basins, PDZ3 samples a range of fully extended conformations with longer distances to both SH3 and GuK (*Figure 6C*).

Examination of structures within the α-basin reveals a broad ensemble of conformations with PDZ3 able to reorient around the HOOK helix and occasionally sample the SH3 RT loop *Kurochkina and Guha, 2013* (*Figure 6—figure supplement 3*). This suggests a fuzzy and dynamic interaction. Examination of α-basin pairwise contacts reveals that basic residues within HOOK interact with negatively charged residues in PDZ3 and are further stabilized by surface-exposed aromatic and hydrophobic residues (*Figure 6E*). Negatively charged residues in α3 and the β2-β3 loop of PDZ3 keep the peptide-binding face oriented towards SH3. Steric clashes would preclude PDZ3 from ligand binding while in some conformations within the α-basin (*Figure 6—figure supplement 4A*). Interestingly, the PDZ3-SH3 linker interacts with the SH3 domain, which helps retain PDZ3 in the α-basin and prevents more

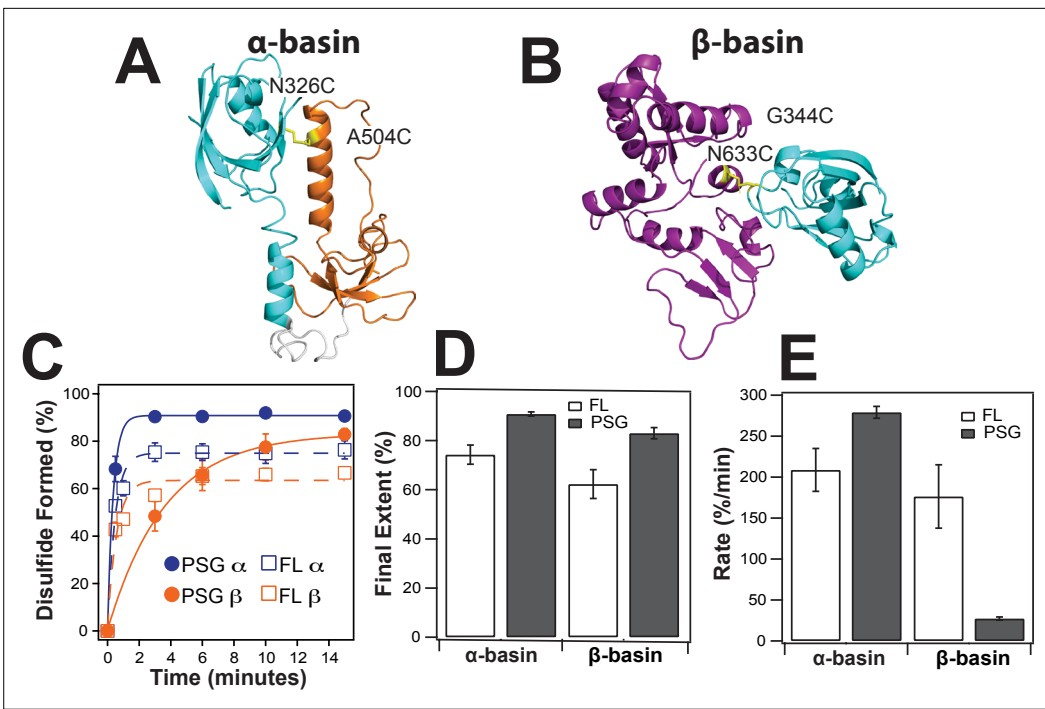

**Figure 7.** Disulfide mapping of the contact interfaces identified from DMD simulations. Cartoon representations showing PDZ3 (cyan), SH3 (orange), and GuK (purple). (**A**) Target model from the α-basin. N326C-A504C disulfide in yellow. (**B**) Target model from the β-basin. G344C-N633C disulfide in yellow. (**C**) Kinetics of disulfide bond formation as measured using non-reducing SDS-PAGE (*Figure 7—figure supplement 1*) showing the α-basin (blue) and the β-basin (orange). Data for full-length PSD-95 are shown as circles with fits as solid lines while data for the PSG truncations is shown as open circles with dashed lines. (**D**) Final extent of disulfide bond formation from single exponential fits to the kinetic data for full-length PSD-95 (white) and the PSG truncation (gray). (**E**) Kinetic rate of disulfide bond formation from single exponential fits to the kinetic data. Error bars represent the SD from three replicate measurements.

The online version of this article includes the following figure supplement(s) for figure 7:

**Figure supplement 1.** Disulfide mapping of the contact interfaces from DMD.

**Figure supplement 2.** Controls for disulfide mapping of the contact interfaces from DMD.

extended conformations. Hydrophobic and electrostatic interactions between the PDZ3-proximal linker (F400, E401, K403, and I404) and SH3 are among the top 50 pairwise residue contacts.

Examination of the β-basin ensemble revealed a more well-defined interaction with PDZ3 binding near the interface of the nucleoside monophosphate (NMP) binding and CORE subdomains of GuK (*Figure 6B*). The interaction does not occlude the canonical peptide binding sites in GuK or PDZ3 (*Figure 6—figure supplement 4B*). Analysis of the β-basin pairwise contacts revealed a well-defined binding site stabilized by hydrophobic and hydrogen bonding interactions between uncharged polar residues (*Figure 6F*) unlike the highly charged interface in the α-basin. Interestingly, the PDZ3-SH3 linker also makes significant pairwise contacts with GuK in this basin. Unexpectedly, the HOOK insertion formed an extended α-helix between residues 491 and 508 in DMD simulations. This is longer than observed in crystal structures (*McGee et al., 2001*; *Tavares et al., 2001*; *Figure 6—figure supplement 4C*). Interestingly, PDZ3 interactions appear to stabilize the helical extension, as is visible in the representative models from each basin (*Figure 6—figure supplement 3A and B*).

## Disulfide mapping the interdomain interfaces

To confirm the PDZ3 interactions with SH3 and GuK, we engineered cysteine residues based on the contact frequency maps and measured the extent and rate of disulfide (DS) bond formation (*Figure 7*). DS formation depends on distance and orientation between cysteines but also contact frequency for dynamic interactions. Thus, the kinetics of bond formation report on structural proximity *Bass et al.,*

*2007b*. We made peripheral cysteine substitutions so as to not disrupt the primary contacts, which are shown using the structure with the lowest root mean square deviation (RMSD) to each basin ensemble. The predominant interface in the α-basin was PDZ3 binding to the HOOK. We probed this basin with the residue pair N326C-A504C, which are only 5.6 Å apart in the α-basin representative (*Figure 7A*). The predominant interface in the β-basin had PDZ3 binding to GuK. We probed this basin with the residue pair G344C-N633C, which are 5.7 Å apart in the β-basin representative (*Figure 7B*).

Increased electrophoretic mobility indicated that DS formation was occurring for all samples (*Figure 7—figure supplement 1*). The data were fit to an exponential function to determine the rate and final extent of DS formation (*Figure 7D*). The α-basin variant showed slightly more DS formation than the β-basin variant in full-length PSD-95 but the rates of DS formation were similar (*Figure 7E & F*). To probe the effects of truncation, we measured DS formation in the PSG truncation. Interestingly, the truncation had opposite effects on the kinetics of DS formation for the two variants. The rate of DS formation for the α-basin variant increased by ~30% while rate of DS formation for the β-basin variant decreased by sixfold.

We also chose two of our existing FRET variants, one with high FRET (P3-S2) and one with mid FRET (P1-G2 *McCann et al., 2012*), and screened them for DS formation (*Figure 6—figure supplement 2*). DMD simulations found that variant P3-S2 occasionally sampled close distances in the α-basin. As predicted, variant P3-S2 formed DS albeit much more slowly than either designed variant. In contrast, our mid FRET variant, which is not involved in contacts in either basin, failed to show any evidence of DS formation even after 1 hr. Thus, DS analysis confirms that the contact interfaces from DMD are sampled in full-length and truncated PSD-95. We also observe significant kinetic differences when PSD-95 is truncated.

## Structural modeling with experimental FRET restraints

To describe the conformation of each limiting FRET state, we used the FRET distances from the global lifetime fits as restraints in rigid body docking. We simulated the accessible volume (AV) for both dyes at each labeling site (*Kalinin et al., 2012*; *Appendix 4—table 1*), which we used as distance loci for each restraint. For each FRET state, we generated 33,000 conformations, which were each scored for agreement between the inter-AV distances and the FRET distances. For state A, the best-fit models showed some divergence with PDZ3 near the HOOK insertion but also extended away from the HOOK in conformations lacking interdomain contacts (*Figure 8A*). The best-fit models for state B are more tightly clustered and position PDZ3 near the NMP subdomain of GuK (*Figure 8B*).

To compare the models from rigid body docking to the DMD ensemble, we calculated the dye accessible volumes for all snapshots structures from the DMD trajectory. For each structure, we plotted the inter-AV distance ($\langle R_{DA} \rangle_{AV}$) against the distance between the labeled domains (*Figure 8—figure supplement 2*). The limiting state distances from the lifetime fits are depicted as orange and purple vertical lines for state A and B, respectively. The experimental limiting states A and B generally fall within the associated α- and β-basins from DMD. However, it was also clear that for some variants, (e.g. P2-G6) the vertical line for state B agrees with both DMD basins. Similarly, the PDZ3-SH3 variants may not fully capture the underlying population distribution. This apparent discrepancy rises from what we call FRET degeneracy in which the interdye distance remains the same despite heterogeneity in the underlying conformational distribution.

To resolve the FRET degeneracy, we introduce the FRET network robustness (FNR) analysis (*Figure 8C & D*). We systematically refit the fluorescence lifetime data using different numbers and combinations of variants from the FRET network. We randomly selected sub-samples with as few as five variants while including each variant in at least three subsets. The distance deviation increased with fewer variants, but the robustness distributions remained centered on the distance from the full global fit (*Figure 8C & D*, histograms). When more than seven variants were used, the robustness deviation was less than 10% regardless of which variants were included in the subset. The SD of the FNR distributions captures the heterogeneity introduced by FRET degeneracy, which can be used (along with AV simulations) to impose new bounds on the structural heterogeneity (*Appendix 4—table 2*).

Using the experimental distance bounds from our FRET network robustness analysis, we screened the 20,871 snapshot structures from DMD simulations to identify all models that are consistent with the experimental FRET data for each limiting state. This identified many more models than is typical

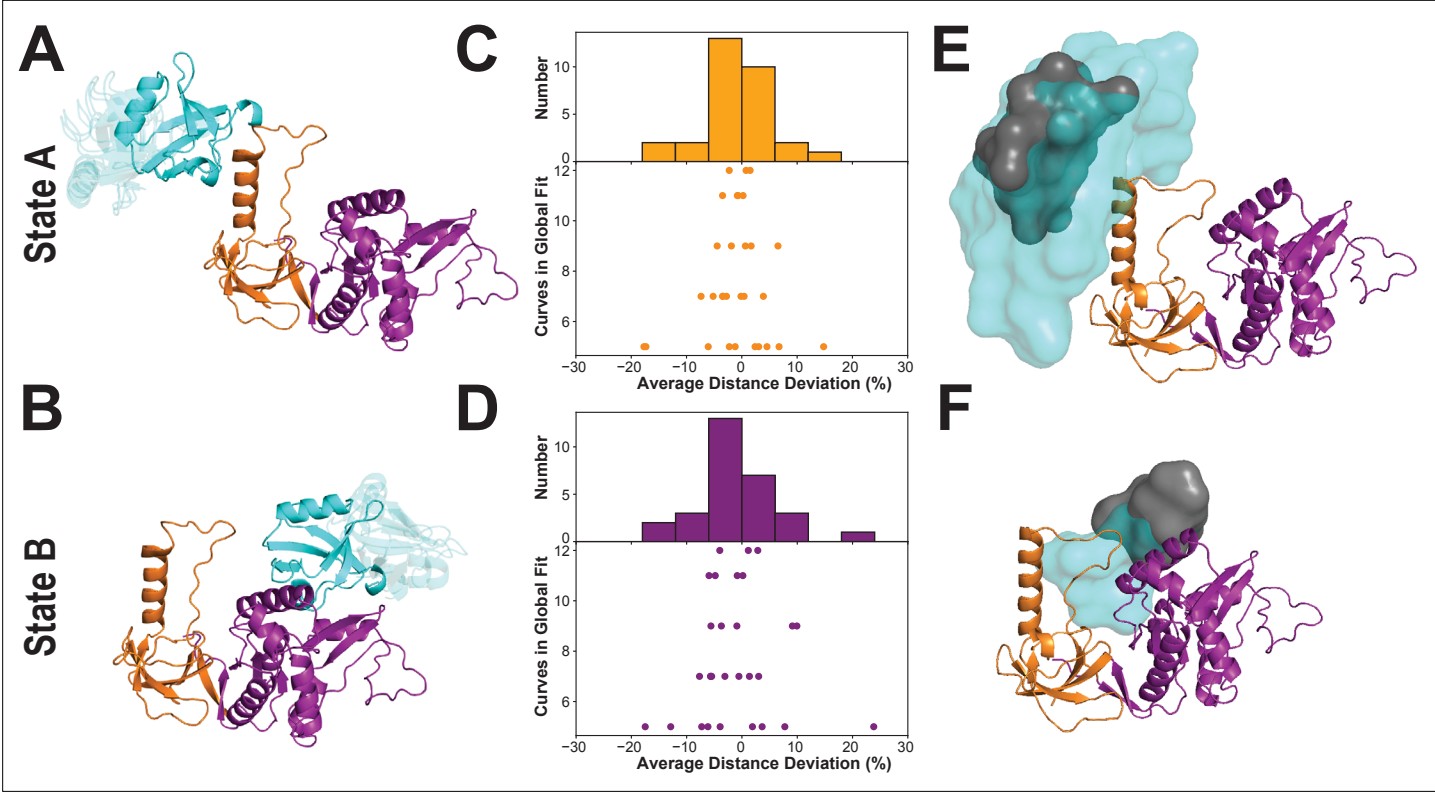

**Figure 8.** Modeling the supertertiary structural ensemble with experimental FRET restraints. (**A & B**) Cartoon representations of the best fit models from rigid body docking of PDZ3 based on the FRET distances with PDZ3 (cyan), SH3 (orange) and GuK (purple) (PDBDEV_00000161). (**A**) Best fit model for full-length PSD-95 in limiting state A. (**B**) Best fit model for limiting state B. (**C & D**) FRET network robustness analysis. We randomly selected sub-samples of the FRET network and repeated a global fit on the reduced FRET network. Histograms show the percent error in distance for all sub-sample fits relative to the global fit. Distance errors from each sub-sample fit are shown beneath the histogram as a function of the number of variants included. Distances and associated widths resulting from subsampled global fitting are summarized in *Appendix 4—table 2*. (**C**) Average distance deviation for state A from global fitting of individual sub-sampled FRET networks. (**D**) Average distance deviation for state B. The histograms show that the distributions are centered on the reported distances regardless of the number of variants. These summary statistics indicate convergence toward the center of the distribution as the number of samples globally fit is increased. (**E & F**) Heterogeneity of the conformational state ensembles based on classification of structures from discrete molecular dynamics (DMD). The position of PDZ3 in each model is represented as a sphere at its center of mass. The gray surfaces represent the 95% confidence intervals for localization of the PDZ3, based on the F-test for the ratio of $\chi^2_r$ for all docking structures relative to the $\chi^2_r$ of the top-ranked structure with nine free parameters (number of distances used for docking of PDZ3). This captures the uncertainty in distances from the global fit but not the full heterogeneity of each basin. The cyan surfaces represent conformational space accessible to PDZ3 within the thresholds from the FRET network robustness analysis shown in panels C and D (*Appendix 4—table 2*) (PDBDEV_00000164). (**E**) Fuzziness of the conformational α-basin related to state A. (**F**). Fuzziness of the conformational β-basin related to state B.

The online version of this article includes the following figure supplement(s) for figure 8:

**Figure supplement 1.** Responsiveness of individual variants to the underlying conformational distribution.

**Figure supplement 2.** Model robustness analysis.

for rigid body docking, from which only a handful of top scoring structures emerge. To condense this large number of models, we represented the PDZ3 center of mass for each compatible model as a sphere and then just display the surface formed by the overlapping spheres. This emphasizes the fuzziness of the experimental states, which encompass a broad conformational ensemble like the basins from DMD. The boundaries of α-basin ensemble are enumerated by the variance of the FNR distributions (*Figure 8C*) but arise from the conformational fuzziness of interdomain interactions between PDZ3 and SH3 (*Figure 8E*). In contrast, the β-basin is not fuzzy (*Figure 8F*). For comparison, we overlay surfaces representing PDZ3 centers of mass from DMD models that fall within the 95% confidence intervals for distances resulting only from the initial global fit of all 12 variants (*Figure 8E and F*; gray spheres). Additionally, we tested whether the intrinsic uncertainty in the individual FRET-derived distances significantly impacted the ensembles α and β (*Figure 8—figure supplement 2*).

We found that the localization of the PDZ3 with respect to the SH3 and GuK was not sensitive to small changes in the mean interdye distances used for FNR that could be explained by the intrinsic uncertainty of the FRET-derived distances.

We used this structural information, along with the dynamic timescales from correlation analysis, to construct a conformational landscape of the PSG supramodule within PSD-95 consistent with DMD and FRET measurements (*Figure 9A*). We performed principal component analysis and projected the basins along the first two principal components (PC1 and PC2) using the simulated interdomain distances and the FRET distances between PDZ3 and SH3-GuK, which were rescaled such that the integrated volume of each basin was equivalent to its experimental population fraction (*Figure 9—figure supplement 1*). PC1 separated the α- and β-basins mostly by interdomain displacement, while PC2 describes the α-basin FRET degeneracy due to domain reorientations within a single basin (*Figure 9B*). Hence, we expand the energy landscape to include degenerate α- and α'basins and link the fFCS relaxation rates directly to the barrier heights for transitions between basins. All structural ensembles in *Figure 8* and kinetic exchange parameters (*Figure 9*) were deposited at the prototype archiving system PDB-Dev with the IDs: PDBDEV_00000161 (*Figure 8A, B*) and PDBDEV_00000164 (*Figure 8E, F*).

## Effect of PDZ3 native context on interactions with neuroligin

Our PSG models include conformations that could impact ligand binding to PDZ3. To test this, we examined the interaction with neuroligin 1a, a key synaptic adhesion protein that interacts with PDZ3 (*Figure 10A*; *Irie et al., 1997*; *Fu et al., 2003*). To compare binding between the truncated PDZ3 and full-length PSD-95, we used a 10 residue C-terminal neuroligin peptide (NL10), which has been reported to bind truncated PDZ3 with an equilibrium dissociation constant ($K_D$) of 10 µM (*Saro et al., 2007*). We used this peptide N-terminally-labeled with fluorescein and measured fluorescence anisotropy. We obtained a $K_D$ of 25±4 µM at pH 6, which is in good agreement given our higher ionic strength. Surprisingly, we were unable to reach saturation when we repeated this measurement at pH 7.4 indicating that the $K_D$ increased to over 200 µM (*Figure 10B*). Thus, neuroligin binding showed a strong pH dependence. Binding to PDZ3 was poor at physiological pH. In contrast to truncated

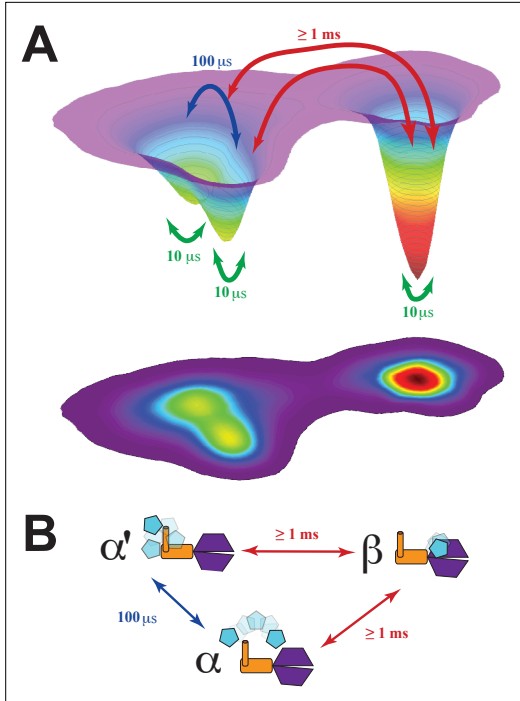

**Figure 9.** Energy landscape for the conformational ensemble of the PSG supramodule within PSD-95. (**A**) Energy landscape surface from principal component analysis (*Figure 9—figure supplement 1*). The basins α and α' correspond to the two sub-basins separated by a small shoulder along principal component 2. The surface was rescaled such that the integrated volumes of basins α and β matched the experimental population fractions from global analysis of the lifetime decays for states A and B, respectively. The transitions and their associated timescales were taken from correlation analysis. The timescale for each transition is indicated by colored arrows. The fastest transitions are associated with local motions, which were not resolved by Förster resonance energy transfer but are inferred from discrete molecular dynamics (DMD) simulations and filtered fluorescence correlation spectroscopy (fFCS). (**B**) Conformational exchange within the energy landscape. Cartoon models for the PSG supramodule in each basin are shown with PDZ3 (cyan), SH3 (orange) and GuK (purple). The timescale for each transition is indicated by colored arrows as in panel A. The movie in *Figure 9—video 1* illustrate the supertertiary dynamics of the PSG supramodule. Structures were sampled from the the ensmbles to PDB-Dev ( PDBDEV_00000164 ).

The online version of this article includes the following video and figure supplement(s) for figure 9:

**Figure supplement 1.** Principal component analysis of COM and AV data.

**Figure 9—video 1.** Illustration of supertertiary dynamics of the PSG supramodule within PSD-95. https://elifesciences.org/articles/77242/ figures#fig9video1

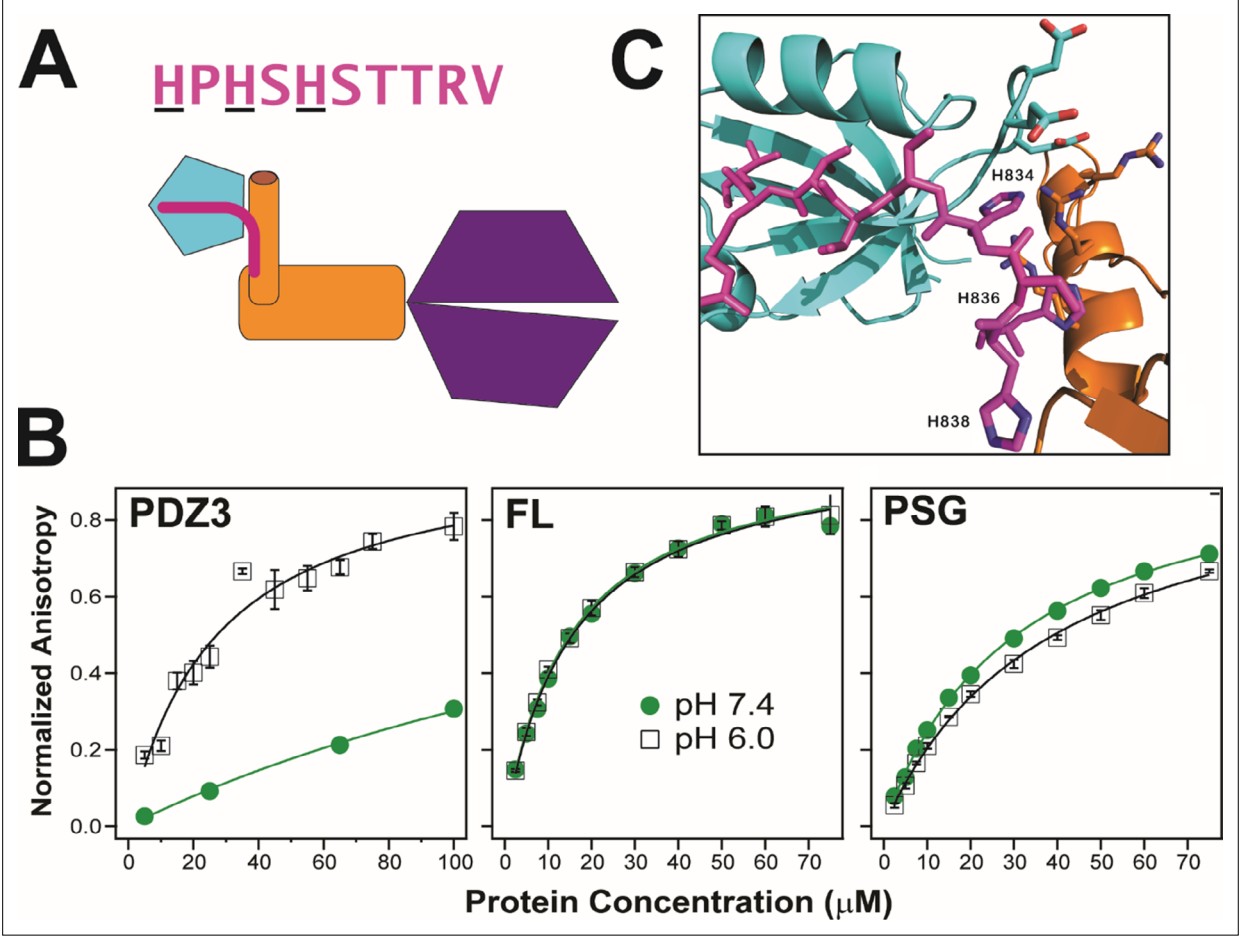

**Figure 10.** Effect of supertertiary environment on neuroligin binding. (**A**) Schematic illustration of the interaction between PSD-95 and the neuroligin peptide (pink). The sequence of the neuroligin C-terminal peptide used in binding experiments is shown with pH-sensitive histidines underlined. (**B**) Representative binding isotherms for the neuroligin peptide binding to truncated PDZ3 (left), full-length PSD-95 (middle) and truncated PSG (right). Shown are data at pH 7.4 (green circles) and pH 6.0 (open squares). Anisotropy values were normalized relative to the final anisotropy taken from non-linear least squares fits (lines). Error bars represent the standard deviation from three replicate measurements. (**C**) Representative conformation from docking of the neuroligin peptide (pink) to PSG in the α–basin from discrete molecular dynamics (DMD) (*Figure 10—figure supplement 1*). The peptide C-terminus interacts with the GLGF motif in PDZ3 (cyan). Charged residue interactions between PDZ3 and SH3 (orange) prevent electrostatic repulsion of histidines that otherwise weakens peptide binding at neutral pH.

The online version of this article includes the following figure supplement(s) for figure 10:

**Figure supplement 1.** Docking of neuroligin to PDZ3 and the PSG supramodule in α-basin.

PDZ3, full length PSD-95 gave a $K_D$ of 15±1 µM at pH 6 and pH 7.4 suggesting the NL10 interaction is enhanced by the supertertiary environment of PDZ3. Next, we examined neuroligin binding to the PSG truncation. We measured a $K_D$ of 39±1 µM at pH 6 suggesting that binding affinity is somewhat reduced in the PSG relative to full length or even the PDZ3 truncation. The PSG showed slightly higher binding affinity at pH 7.4 with a $K_D$ of 31±2 µM in stark contrast to the reduced binding affinity of the truncated PDZ3. Thus, effects of truncation on supertertiary structure and dynamics impact NL10 binding.

To understand this phenomenon, we performed docking of NL10 to truncated PDZ3 at pH 6. We identified electrostatic interactions between protonated histidines in NL10 and acidic residues in the PDZ3 β2-β3 loop (*Figure 10—figure supplement 1*). At pH 7.4, unprotonated histidines are incapable of interacting with negatively charged residues, which explains the pH sensitivity. The resulting desolvation penalty of the unpaired acidic residues would reduce the binding affinity. Some α-basin conformations block peptide binding due to steric clashes. However, docking of NL10 to α-basin structures identified multiple PSG conformations where the acidic residues in PDZ3 interacted with

basic residues in HOOK without steric clashes for NL10 (*Figure 10C*). Salt-bridges have a smaller desolvation penalty than unsatisfied charges, which explains why the peptide binds full-length PSD-95 (and PSG) better than truncated PDZ3 at pH7.4. Thus, the supertertiary context of PDZ3 enables recognition of a critical physiological ligand.

## Discussion

PSD-95 is a scaffold protein at excitatory synapses that forms a crucial link between neurotransmitter release and detection pathways (*Won et al., 2017*). PSD-95 brings together different binding partners from oligomeric transmembrane proteins to soluble enzymes. Proteomic analysis of PSD-95 complexes purified from mouse brain identified 301 different proteins (*Fernández, 2009*). In many cases, the binding partners are larger than the scaffolding domains with which they interact. Kinetic analysis showed that higher-order interactions between proteins bound to PSD-95, play a role in scaffolding activity (*McCann et al., 2014*). The assembly of multi-protein complexes poses steric challenges. Thus, the dynamic supertertiary structure must play a role in accommodating multiple partners.

Here we used our combination of experimental and computational methods for dynamic structural biology (*Figure 2*) to describe the supertertiary structure and dynamics of the conserved PSG supramodule within full-length PSD-95. Multiparameter fluorescence analysis revealed a complex and dynamic conformational landscape. DMD and modeling based on FRET distances were in excellent agreement on the location of PDZ3 within two non-overlapping basins. Both DMD and FRET agree that the α-basin was degenerate due to the fuzzy interface allowing PDZ3 reorientation. Crucial for the experimental analysis was pulsed interleaved excitation combined with filtering to generate a subensemble containing molecules with an active donor and acceptor pair, which is also a benefit of smTIRF. Correlation analysis of this subensemble was able to assign dynamic timescales to the conformational transitions, which we summarized as an energy landscape (*Figure 9A*). Within each basin, each domain undergoes rapid local motion. Reorientation of PDZ3 and sampling of extended states is slower while basin exchange takes place on timescales approaching seconds, as captured by smTIRF.

Our models were in excellent agreement with SAXS and NMR experiments, which identified multiple conformations with PDZ3 localized predominantly near the HOOK insertion and suggested involvement of residues around the PDZ3 peptide binding pocket (*Zhang et al., 2013*). However, the previous Rosetta modeling of linker suggested the PDZ3-SH3 linker bridged the interaction while DMD simulations suggested the PDZ3-SH3 interaction was direct. We note that neither the current nor previous study used experimental data in the refinement of linker models. In DMD simulations, the linker interacts with SH3 and GuK to help retain PDZ3. This linker is almost 100% conserved in PSD-95 homologues from humans to *Drosophila*, which would be unusual unless involved in protein interactions since disordered linkers usually show reduced sequence conservation (*Chen et al., 2006*). Neither basin relied on canonical ligand binding modes for the primary interaction as suggested by comparative patch analysis (*Korkin et al., 2006*). Additionally, neither basin corresponded to the position for PDZ3 that we previously identified as the time-averaged location using smTIRF (*McCann et al., 2012*). Our earlier models were generated using a similar rigid body docking approach but our experimental data lacked the time resolution necessary to identify both conformations with the complex conformational landscape. Modeling the PSG as a single conformation did not capture either basin model.

The crystal structure of the PSG supramodule from ZO-1, a MAGuK protein found at intercellular tight junctions, revealed a lack of interdomain interactions (*Nomme et al., 2011*). The PDZ3-SH3 linker in ZO-1 is much shorter than PSD-95, which may prevent access to these binding sites. The ZO-1 HOOK insertion also lacks sequence similarity to PSD-95 (*Nomme et al., 2011*). In contrast, the basic residues in HOOK and acidic residues in PDZ3 are conserved among synaptic MAGuK homologs PSD-93, SAP97, and SAP102 as are most pairwise contacts in the β-basin (e.g., F339-Q594, L342-Q603). This explains why the supertertiary landscape is conserved in these homologues as suggested by previous smTIRF measurements (*McCann et al., 2012*).

The importance of supertertiary interactions on scaffolding activity is emphasized by the enhanced binding of neuroligin to full-length PSD-95. Other PDZ3 ligands (e.g. CRIPT, stargazin and synGAP) contain lysines and arginines near the canonical PDZ motif while neuroligin contains histidines, which explains why NL10 has pH sensitivity. In the fuzzy α-basin, interactions with positive charges in the HOOK satisfy negatively charged residues in PDZ3, which facilitates uncharged peptide binding.

Additionally, fuzziness increases the favorability of the α-basin by allowing PDZ3 to reorient to satisfy multiple interactions. These supertertiary interactions also affect other ligands. Measurements of CRIPT binding to PDZ3 suggested that PDZ3 adopted two interconverting conformational states in the PSG with different kinetics but the same equilibrium affinity (*Laursen et al., 2020*). Thus, the supertertiary context of PDZ3 is necessary to overcome the repulsive interactions that prevent neuroligin binding to the truncated domain.

## Methods

### Protein expression and purification

The full-length PSD-95 from *Rattus norvegicus,* the PSG truncation (residues 303–274) and truncated PDZ3 (303–415) were expressed in the Rosetta 2 strain of *E. coli* (EMD Millipore) and purified by a combination of Ni-affinity, ion exchange and size exclusion chromatography as previously described for these exact protein constructs (*McCann et al., 2011*). Each variant was expressed and purified separately generating enough material for separate measurements by smTIRF and MFD. The elution conditions for each variant were similar during anion exchange and size exclusion chromatography suggesting a lack of gross perturbations of protein folding and conformation. This should not be surprising as each variant has only two conservative amino acid substitutions in surface exposed residues as identified from existing protein structures for each domain. Furthermore, these labeling variants were used previously, which included control photophysical measurements of the labeled proteins along with analytical size exclusion of the protein constructs (*McCann et al., 2012*). For smTIRF, proteins were labeled Alexa Fluor 555 $C_2$ maleimide and Alexa Fluor 647 $C_2$ maleimide at an equimolar ratio. For MFD, proteins were first labeled with a 1:2 ratio of Alexa 488 $C_5$ maleimide for 1 hr at 4°C followed by addition of a 5:1 molar ratio of Alexa Fluor 647 $C_2$ maleimide, which was reacted overnight at 4°C. Unconjugated dye was removed by desalting with Sephadex G50 (GE Healthcare) followed by dialysis.

### Single-molecule total internal reflection fluorescence (TIRF)

Fluorescently labeled PSD-95 was encapsulated in 100 nm liposomes supplemented with 0.1% biotinylated phosphoethanolamine (Avanti Polar Lipids, Alabaster, AL). Unencapsulated protein was removed by desalting on Sepharose CL-4B (GE Healthcare). Liposomes were attached via streptavidin to a quartz slide passivated with biotinylated-BSA. FRET data was collected at 10 frames/second using an Andor iXon EMCCD camera (Andor Technologies, Belfast, UK). All smFRET measurements were performed in 50 mm Tris 150 mM NaCl pH 7.5 supplemented with 1 mM cycooctatetraene, 0.5% w/v glucose, 7.5 units/mL glucose oxidase and 100 units/mL catalase. Alternating illumination using 640 and 532 nm lasers allowed for the identification of molecules containing one donor and one acceptor. Microscopy data was analyzed using MATLAB to perform the mathematics needed to correlate donor and acceptor images, extract single-molecule intensity time traces and calculate FRET efficiency. Per molecule gamma normalization based upon acceptor photobleaching events was used to correct for distortions in the measured intensities between the donor and acceptor channels (*McCann et al., 2010*).

### Multiparameter fluorescence detection

Two experimental setups were used for confocal measurements to obtain MFD data (*Appendix 1—table 1*). Both setups utilized pulsed interleaved excitation (PIE) to alternately excite donor and acceptor fluorophores directly (*Kudryavtsev et al., 2012*). Emission was separated into parallel and perpendicular polarization components at two different spectral windows using band pass filters as described in supplementary methods. Labeled samples were diluted in 50 mM sodium phosphate, pH 7.5, 150 mM NaCl, 40 µM TROLOX, which had been charcoal filtered to remove residual impurities. We used pM concentrations such that we observed ~1 molecule per second in the confocal volume. Samples were measured in NUNC chambers (Lab-Tek, Thermo Scientific, Germany) that were pre-coated with a solution of 0.01% Tween 20 (Thermo Scientific) in water for 30 min to minimize surface adsorption. We obtained the instrument response function (IRF) using water. Protein-free buffer was used for background subtraction. Calibration experiments and data collection were as previously reported (*Ma et al., 2017*). Burst selection was performed using all-photon, inter-photon arrival time

traces to identify single molecules. Burst selection criteria were set such that each burst contained a minimum of 40 photons summed amongst all detection channels, with an inter-photon arrival time cutoff set to the mean minus one standard deviation, calculated across the entire time trace. The donor fluorescence lifetime and the intensity-based FRET efficiency were calculated for each burst using a maximum-likelihood estimation algorithm (*Yanez Orozco et al., 2018*; *Maus et al., 2001*). To ensure both fluorophores were present in each select burst used in MFD histograms, we used cutoff values for (1) the difference between observed burst duration in green and red channels under direct excitation of the corresponding fluorophores ($|T_{GG}-T_{RR}|<1ms$) and (2) the observed FRET stoichiometry ($0.3<S_{PIE} < 0.7$) (*Yanez Orozco et al., 2018*).

## Structure generation and system setup

The PDZ3 and SH3-GuK domains of the PSG core were reconstructed using homology modeling of I-TASSER (Iterative Threading ASSEmbly Refinement) hierarchical approach using multiple threading alignments and iterative structural assembly simulations. The I-TASSER method involves four consecutive stages: (a) template identification or threading by LOMETS, (b) structure assembly by replica-exchange Monte Carlo simulations, (c) model selection and structure refinement using REMO and FG-MD, and (d) structure-based function annotation using COFACTOR (*Roy et al., 2011*; *Roy et al., 2010*). The individual PDZ3, SH3-GuK domains were positioned randomly and our in-house loop reconstruction program 'medusa-loop' was utilized to connect the individual domains via flexible linkers to reconstruct the PSG core (*Figure 6—figure supplement 1*). The backbone of the GuK domain was kept static while the SH3 and PDZ3 domains were kept flexible to allow adequate sampling of the conformational landscape. The Gō constraints were imposed on the internal residues within PDZ3 (308–405), the internal residues within SH3 (431–531) domains and the β strand F of GuK (residues 710–724) that folds back to form an antiparallel β sheet with β strand E of the SH3 domain. The implementation of Gō-like constraints, based on inter-residue contacts, can provide a reliable model consistent with the experimentally-established native structure of PDZ3 and SH3-GuK.

## Replica exchange DMD simulations

Replica exchange DMD (rxDMD) simulations were performed with 18 neighboring replicas in the temperature range of 275–360 K to efficiently sample the conformational free energy landscape. In the rxDMD scheme, temperature was exchanged between replicas at periodic time intervals, according to the Metropolis criterion, to overcome any local energy barriers that may limit efficient sampling of the conformational states. Production runs followed 3000 timestep energy minimization runs, where a DMD timestep corresponded to ~50 fs. The conditions for replica exchange were checked every 1000-time units and the frames were saved every 100-time units. Anderson's thermostat was used to maintain temperature at 300 K in all simulations and the heat exchange factor was set to 0.1. Each replica simulation lasted for 660 ns that resulted in a total simulation time of ~11.9 μs.

The thermodynamics of the PSG core were computed using the Weighted Histogram Analysis Method (WHAM) (*Feig et al., 2004*; *Kumar et al., 1992*). The time evolution of the $R_g$ for the 18 simulated replicas is provided in *Figure 6—figure supplement 2*. We observed dynamic, spontaneous conformational exchange in each of the replicas (*Figure 6—figure supplement 2*), highlighting the efficient sampling from exchanging temperature between replicate simulations. The last 400 ns trajectories from all 18 replicas were used for the WHAM analysis. The potential of mean force (PMF)–that is the effective free energy–was estimated according to

$$E(R_{PMF}) = -K_B T \, ln(p(R_{PMF})), \tag{1}$$

where $R_{PMF}$ is the multiple dimensional parameter such inter-domain distances, $p(R_{PMF})$ is the probability density derived from WHAM, $T$ is the temperature of interest and $k_B$ is the Boltzmann constant. Our 2D free energy landscape was computed as a function of interdomain distances between PDZ3, SH3$_{Hook}$, SH3$_{barrel}$, and GuK. The free energy contours are scaled in units of kcal/mol.

The hierarchical clustering program (https://www.compbio.dundee.ac.uk/downloads/oc) was utilized to group similar conformations of the PSG core. Depending on the pair-wise RMSD matrix, the clustering algorithm iteratively combines nearby clusters. The 'cluster distance' was determined based on all pairwise distances between elements of corresponding clusters. We used the mean of all

values to compute the distance between two clusters, and the centroid structure of each cluster was chosen with the smallest average distance to other elements in the cluster.

The disulfide bond modeling of the interdomain contacts in PSG was performed for the two basins (α and β) in the free energy landscape. From the structural ensembles corresponding to the reaction coordinates in the residual contact frequency map, we ranked contact residue pairs and selected the top three pairs near the interface that did not involve the hydrophobic and salt-bridge interactions. The selected pairs were situated at the peripheral sites of the interdomain interface, and the corresponding residues were either polar or weakly hydrophobic in nature (*Bass et al., 2007b*).

## Disulfide mapping in PSD-95

All samples were prereduced with 5 mM DTT for 1 hr followed by desalting using a PD10 column (GE Healthcare) into 20 mM Tris pH 7.4, 150 mM NaCl. Disulfide oxidation reactions were performed at 25°C with 2 µM protein concentration for the PSG fragment or 0.5 µM for full-length PSD-95. Disulfide formation was initiated by the addition of 0.5 mM $CuSO_4$ and 1.75 mM 1, 10-phenanthroline (*Bass et al., 2007a*). Time points were quenched by adding 40 mM N-ethylmaleimide and 10 mM EDTA. Samples were boiled in non-reducing Laemmli sample buffer and run on 10 or 5% SDS-PAGE for PSG and full-length, respectively. All experiments were carried out in triplicate. Intensities for native and shifted bands were measured in ImageJ. Percentages of disulfide formation were calculated for each time point and corrected for the presence of higher order oligomers. Each reaction was well fit to a single exponential function to obtain the initial and final extent of disulfide formation along with the reaction rate for each mutant. Replicates were analyzed separately to obtain the average and standard error of measurement (SEM) as well as to estimate the error in the fitted parameters.

## Neuroligin peptide binding

Binding experiments used protein constructs described above with a 10-residue synthetic peptide corresponding to the C-terminal 10 residues from rat neuroligin 1 a (HPHSHSTTRV), which was synthesized with an N-terminal fluorescein label (5-FAM, GenScript USA Inc Piscataway, NJ). Fluorescence polarization measurements were carried out in black 96-well plates measured on a Wallac Victor 2 Plate Reader (Perkin Elmer). For all measurements peptide was held at concentrations of 50 nm (or 100 nM) while the protein concentrations ranged from 1 to 100 µM. Measurements at pH 7.4 were made in 20 mM Tris, 150 mM NaCl, 1 mM DTT, and 1 mM EDTA. Measurements at pH 6.0, were made in 20 mM MES, 150 mM NaCl, 1 mM DTT, and 1 mM EDTA. Four readings were taken for each well, then averaged. All experiments were carried out in triplicate and fitted with the Hill Equation for a single site binding site model.

## Code availability

MFD is made available at http://www.mpc.hhu.de/en/software. DMD simulation engine is available at http://www.moleculesinaction.com. MATLAB scripts used to analyze smTIRF are included as *Source code 1*.

## Acknowledgements

This work was supported by NIH (MH081923 to MEB, 1P20GM130451 to HS, and GM119691 to FD), NSF (CAREER MCB 1749778 to HS and CAREER CBET-1553945 to FD); and ERC (AdG2014 hybridFRET # 671208) to CS and HHU Connect to GH, HS, and CS. We thank Markus Seeliger (NIH S10OD028478) and Aziz Rangwala for technical assistance with peptide binding experiments. We thank Christian Hanke (HHU, Germany) and Brinda Vallat (RCSB at Rutgers, NSF awards DBI-2112966, DBI-2112967, and DBI-2112968) for helping with submission of the structure ensembles at PDB-Dev.

## Additional information

### Funding

| Funder | Grant reference number | Author |
| --- | --- | --- |
| National Institute of Mental Health | MH081923 | Mark E Bowen |
| National Institute of General Medical Sciences | GM130451 | Hugo Sanabria |
| National Institute of General Medical Sciences | GM119691 | Feng Ding |
| National Science Foundation | MCB1749778 | Hugo Sanabria |
| National Science Foundation | CBET1553945 | Feng Ding |
| European Research Council | 671208 | Claus AM Seidel |

The funders had no role in study design, data collection and interpretation, or the decision to submit the work for publication.

### Author contributions

George L Hamilton, Data curation, Software, Formal analysis, Funding acquisition, Investigation, Visualization, Methodology, Writing – original draft, Writing – review and editing; Nabanita Saikia, Formal analysis, Investigation, Visualization, Writing – original draft; Sujit Basak, Yan Hao, Formal analysis, Investigation; Franceine S Welcome, Jakub Kubiak, Formal analysis, Investigation, Methodology; Fang Wu, Resources, Investigation, Methodology; Changcheng Zhang, Formal analysis, Investigation, Writing – review and editing; Claus AM Seidel, Conceptualization, Resources, Software, Supervision, Writing – review and editing; Feng Ding, Conceptualization, Resources, Data curation, Formal analysis, Supervision, Validation, Methodology, Writing – original draft, Writing – review and editing; Hugo Sanabria, Conceptualization, Resources, Software, Supervision, Investigation, Visualization, Writing – original draft, Writing – review and editing; Mark E Bowen, Conceptualization, Resources, Formal analysis, Supervision, Funding acquisition, Validation, Visualization, Writing – original draft, Project administration, Writing – review and editing

### Author ORCIDs

George L Hamilton http://orcid.org/0000-0002-2519-3641
Claus AM Seidel http://orcid.org/0000-0002-5171-149X
Feng Ding http://orcid.org/0000-0003-1850-6336
Hugo Sanabria http://orcid.org/0000-0001-7068-6827
Mark E Bowen http://orcid.org/0000-0002-9525-6986

### Decision letter and Author response

Decision letter https://doi.org/10.7554/eLife.77242.sa1
Author response https://doi.org/10.7554/eLife.77242.sa2

## Additional files

### Supplementary files

• Transparent reporting form

• Source code 1. This zip archive contains the MATLAB scripts used to extract FRET efficiency from emCCD camera data collected with smTIRF as shown in *Figure 4*. Included are scripts to align 2 channel FRET images, extract time traces for intensity maxima, manually select single molecule traces and calculate FRET efficiency from selected molecules. A README file is added to aid in implementation of the scripts with MATLAB.

## Data availability

Datasets of FRET values from smTIRF have been uploaded as Figure 4-Source Data 1. The MATLAB scripts used to analyze smTIRF data have been uploaded as Source Code 1. Datasets from Confocal Microscopy with Multiparameter Fluorescence Detection (Raw, MFD Bursts, TCSPC, PDA, and FCS), along with structures used in generating simulated distances and DMD screening, are available at Zenodo (DOI: https://doi.org/10.5281/zenodo.6983428). Datasets from Discrete Molecular Dynamics are available at https://dlab.clemson.edu/research/PSD95-PSG/. FRET-restrained integrative structure models of the PSG supramodule from PSD-95 were deposited to PDB-Dev PDB-Dev ID: PDBDEV_00000161 (generated by docking) and PDBDEV_00000164 (generated by Screening of FNR Ensembles) using the FLR-dictionary extension (developed by PDB and the Seidel group) available on the IHM working group GitHub site (https://github.com/ihmwg/FLR-dictionary). Further data sets generated during and/or analyzed during the current study are available from the corresponding authors on reasonable request.

The following dataset was generated:

| Author(s) | Year | Dataset title | Dataset URL | Database and Identifier |
| --- | --- | --- | --- | --- |
| Hamilton G, Saikia N, Basak S, Welcome FS, Wu F, Kubiak J, Changcheng Z, Hao Y, Seidel CAM, Claus AM, Ding F, Sanabria H, Bowen ME | 2022 | Datasets from Confocal Microscopy with Multiparameter Fluorescence Detection (Raw, MFD Bursts, TCSPC, PDA, and FCS), along with structures used in generating simulated distances and DMD screening | https://doi.org/10.5281/zenodo.6983428 | Zenodo, 10.5281/zenodo.7079661 |

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

## Appendix 1

**Appendix 1—table 1.** Details for MFD instruments used in this study.
Correction parameters for MFD that are specified for each experimental setup. The setup used to measure each variant is indicated in *Appendix 1—table 2*. HHU stands for Heinrich Heine University.

| Parameter | Setup 1 (Clemson) | Setup 2 (HHU) |
|---|---|---|
| Det. Efficiency Ratio (G/R) | 3.7 | 0.8 |
| Green Power (485 nm) | 80 µW | 60 µW |
| Red Power (640 nm) | 32 µW | 10 µW |
| PIE Repetition Rate | 20 MHz | 32 MHz |
| Repetition Time | 50.00 ns | 31.25 ns |
| Direct Acceptor Excitation, δ | 2.2 | 1.3 |
| Spectral Crosstalk, G→R, α | 1.7 | 1.7 |

**Appendix 1—table 2.** Correction parameters for multiparameter fluorescence detection.
Correction parameters determined for each FRET variant based on buffer measurements for backgrounds, donor-only and directly-excited acceptor fluorescence for quantum yields ($QY_G$, $QY_R$), and standard stock fluorophore solutions for G-factors. Parameters constant for all samples measured on a given setup are summarized at the end. Details of variants can be found in *Figure 1* and *Table 1*. HHU stands for Heinrich Heine University.

| FL Variant | $QY_G$ | $QY_R$ | $BG_{G(D)}$ (kHz) | $BG_{R(D)}$ (kHz) | $BG_{R(A)}$ (kHz) | G-factor | Setup |
|---|---|---|---|---|---|---|---|
| P2-G6 | 0.72 | 0.26 | 0.85 | 0.45 | 0.30 | 1.06 | HHU |
| P3-G4 | 0.75 | 0.53 | 0.54 | 0.21 | 0.38 | 1.06 | HHU |
| P2-G1 | 0.72 | 0.38 | 0.53 | 0.18 | 0.17 | 0.83 | Clemson |
| P3-G5 | 0.77 | 0.46 | 0.51 | 0.16 | 0.17 | 0.83 | Clemson |
| P1-G3 | 0.71 | 0.37 | 0.38 | 0.14 | 0.15 | 0.83 | Clemson |
| P1-G4 | 0.76 | 0.39 | 0.41 | 0.15 | 0.15 | 0.83 | Clemson |
| P1-G2 | 0.75 | 0.37 | 0.54 | 0.20 | 0.19 | 0.83 | Clemson |
| P3-S2 | 0.81 | 0.43 | 0.50 | 0.16 | 0.17 | 0.83 | Clemson |
| P2-S2 | 0.68 | 0.26 | 0.78 | 0.45 | 0.36 | 1.06 | HHU |
| S2-G6 | 0.70 | 0.26 | 0.74 | 0.39 | 0.29 | 1.06 | HHU |
| S3-G1 | 0.71 | 0.44 | 0.51 | 0.26 | 0.40 | 1.06 | HHU |
| S1-G1 | 0.73 | 0.49 | 0.51 | 0.26 | 0.40 | 1.06 | HHU |

| PSG Variant | $QY_G$ | $QY_R$ | $BG_{G(D)}$ (kHz) | $BG_{R(D)}$ (kHz) | $BG_{R(A)}$ (kHz) | G-factor | Setup |
|---|---|---|---|---|---|---|---|
| P2-G1 | 0.72 | 0.31 | 0.56 | 0.18 | 0.17 | 0.83 | Clemson |
| P3-G5 | 0.72 | 0.35 | 0.51 | 0.16 | 0.17 | 0.83 | Clemson |
| P1-G3 | 0.74 | 0.40 | 0.38 | 0.14 | 0.15 | 0.83 | Clemson |
| P1-G4 | 0.69 | 0.39 | 0.41 | 0.15 | 0.15 | 0.83 | Clemson |
| P1-G2 | 0.75 | 0.35 | 0.53 | 0.19 | 0.17 | 0.83 | Clemson |
| P3-S2 | 0.77 | 0.38 | 0.50 | 0.16 | 0.17 | 0.83 | Clemson |

**Appendix 1—table 3.** Fit parameters from seTCSPC using a Two State Model.
Species fluorescent lifetimes were used to estimate effective fluorescence-averaged lifetimes accounting for fluorophore linker dynamics. Details on model function and definition of parameters can be found in Materials and Methods. Information about variants is found in *Figure 1* and *Table 1*.

| FL Variant | Donor Fraction | Scattering Amplitude | $\tau_{D(0)}$ | $\langle\kappa^2\rangle$ | $r_{D|D,\infty}$ | $r_{A|A,\infty}$ | $r_{A|D,\infty}$ | Species Fraction $x_I$ | Species Fraction $x_{II}$ | $\tau_{D(A),xI}$ | $\tau_{D(A),xII}$ | $\tau_{D(A),fI}$ | $\tau_{D(A),fII}$ | $\chi^2_{r,seTCSPC}$ |
|---|---|---|---|---|---|---|---|---|---|---|---|---|---|---|
| P2-G6 | 0.02 | 0.07 | 3.60 | 0.72 | .09 | .14 | .07 | 0.46 | 0.54 | 2.73 | 0.39 | 2.78 | 0.65 | 2.54 |
| P3-G4 | 0.00 | 0.17 | 3.76 | 0.82 | .18 | .14 | .08 | 0.46 | 0.54 | 2.67 | 0.53 | 2.75 | 0.83 | 1.49 |
| P2-G1 | 0.01 | 0.04 | 3.59 | 0.85 | .14 | .19 | <.01 | 0.46 | 0.54 | 3.41 | 0.46 | 3.41 | 0.74 | 1.22 |
| P3-G5 | 0.15 | 0.12 | 3.86 | 0.85 | .14 | .21 | .02 | 0.46 | 0.54 | 2.45 | 0.33 | 2.57 | 0.59 | 1.13 |
| P1-G3 | 0.39 | 0.09 | 3.56 | 0.83 | .19 | .13 | .01 | 0.46 | 0.54 | 3.31 | 0.97 | 3.31 | 1.25 | 2.12 |
| P1-G4 | 0.11 | 0.05 | 3.79 | 0.89 | .15 | .22 | .08 | 0.46 | 0.54 | 3.58 | 1.13 | 3.58 | 1.41 | 2.11 |
| P1-G2 | 0.00 | 0.05 | 3.77 | 0.83 | .12 | .20 | .04 | 0.46 | 0.54 | 3.44 | 0.15 | 3.45 | 0.34 | 1.15 |
| P3-S2 | 0.02 | 0.0 | 4.04 | 0.88 | .19 | .25 | .02 | 0.46 | 0.54 | 0.28 | 2.09 | 0.37 | 2.27 | 0.45 |
| P2-S2 | 0.00 | 0.20 | 3.38 | 0.71 | .04 | .20 | .04 | 0.46 | 0.54 | 0.41 | 2.03 | 0.67 | 2.14 | 1.96 |
| S2-G6 | 0.00 | 0.16 | 3.50 | 0.72 | .11 | .13 | <.01 | 0.46 | 0.54 | 0.33 | 1.80 | 0.58 | 1.97 | 0.92 |
| S3-G1 | 0.00 | 0.20 | 3.56 | 0.77 | .14 | .05 | .01 | 0.46 | 0.54 | 0.80 | 2.95 | 1.09 | 2.98 | 1.84 |
| S1-G1 | 0.00 | 0.12 | 3.64 | 0.79 | .18 | .14 | .09 | 0.46 | 0.54 | 2.96 | 0.76 | 2.99 | 1.06 | 1.61 |

| PSG Variant | Donor Fraction | Scattering Amplitude | $\tau_{D(0)}$ | $\kappa^2$ | $r_{D|D,\infty}$ | $r_{A|A,\infty}$ | $r_{A|D,\infty}$ | Species Fraction $x_I$ | Species Fraction $x_{II}$ | $\tau_{D(A),xI}$ | $\tau_{D(A),xII}$ | $\tau_{D(A),fI}$ | $\tau_{D(A),fII}$ | $\chi^2_{r,seTCSPC}$ |
|---|---|---|---|---|---|---|---|---|---|---|---|---|---|---|
| P2-G1 | 0.00 | 0.09 | 3.60 | 0.79 | .14 | .08 | .07 | 0.52 | 0.49 | 3.01 | 0.65 | 3.03 | 0.94 | 0.94 |
| P3-G5 | 0.04 | 0.10 | 3.59 | 0.85 | .14 | .25 | .02 | 0.52 | 0.49 | 1.93 | 0.36 | 2.09 | 0.62 | 0.85 |
| P1-G3 | 0.11 | 0.06 | 3.72 | 0.84 | .19 | .16 | .01 | 0.52 | 0.49 | 3.57 | 0.92 | 3.57 | 1.22 | 2.14 |
| P1-G4 | 0.05 | 0.06 | 3.46 | 0.87 | .14 | .25 | .07 | 0.52 | 0.49 | 3.38 | 0.66 | 3.38 | 0.94 | 1.37 |
| P1-G2 | 0.00 | 0.06 | 3.77 | 0.83 | .14 | .18 | .04 | 0.52 | 0.49 | 3.12 | 0.23 | 3.14 | 0.45 | 1.64 |
| P3-S2 | 0.05 | 0.07 | 3.87 | 0.86 | .14 | .25 | .04 | 0.52 | 0.49 | 0.47 | 2.52 | 0.77 | 2.62 | 0.72 |

**Appendix 1—table 4.** Interdye distances from the global fit of seTCSPC decays.
Distances resulting from seTCSPC fits. Model details are in Materials and Methods. All R0 for distance calculations taken as 52 Å, state widths set to 6 Å. Model fit parameters can be found in *Appendix 1—table 3*. Details of variants can be found in *Figure 1* and *Table 1*. Uncertainties corresponding to 95% confidence intervals were estimated using the F-test for the ratio of $\chi^2_{r,seTCSPC}$ of the final fit to the $\chi^2_{r,seTCSPC}$ under variation of each parameter independently. The number of degrees of freedom was 2663 for all curves (number of data points-number of parameters).

| FL Variant | $\langle R_{DA,A}\rangle$ (Å) | $\langle R_{DA,B}\rangle$ (Å) |
|---|---|---|
| P2-G6 | 64.0±2.0 | 36.6±4.4 |
| P3-G4 | 60.4±2.9 | 38.5±1.1 |
| P2-G1 | 84.7±7.0 | 37.8±3.8 |
| P3-G5 | 57.1±3.2 | 35.0±6.0 |
| P1-G3 | 79.8±9.8 | 44.2±8.3 |
| P1-G4 | 83.3±5.2 | 45.1±5.8 |
| P1-G2 | 76.9±7.4 | 30.6±4.5 |
| P3-S2 | 33.8±6.0 | 52.6±2.1 |
| P2-S2 | 37.4±3.5 | 55.6±1.6 |
| S2-G6 | 35.6±3.4 | 52.5±1.4 |
| S3-G1 | 42.3±2.8 | 67.8±4.2 |
| S1-G1 | 66.5±2.9 | 41.7±2.1 |
| FL Fraction | 46.1% | 53.9% |

| PSG Variant | $\langle R_{DA,A}\rangle$ (Å) | $\langle R_{DA,B}\rangle$ (Å) |
|---|---|---|
| P2-G1 | 68.2±6.2 | 40.4±7.6 |

*Continued on next page*

*Continued*

| PSG Variant | $\langle R_{DA,A}\rangle$ (Å) | $\langle R_{DA,B}\rangle$ (Å) |
|---|---|---|
| P3-G5 | 53.3±3.0 | 36.0±5.5 |
| P1-G3 | 89.0±10.5 | 43.2±4.3 |
| P1-G4 | 96.5±12.0 | 40.8±2.0 |
| P1-G2 | 67.5±2.1 | 32.9±3.5 |
| P3-S2 | 37.4±3.9 | 57.7±5.7 |
| PSG Fraction | 51.8% | 48.2% |

**Appendix 1—table 5.** Parameters for calculation of static FRET-lines in MFD plots.
The static FRET-lines are calculated using these parameters and shown for visual reference in
MFD plots in *Figure 3* and *Figure 4—figure supplement 2*. FRET-lines are corrected for dynamic
averaging due to fluorophore linker dynamics via polynomials relating the species fluorescence
lifetimes, $\tau_{D(A),x}$, from seTCSPC fits and linker-movement corrected apparent fluorescence-averaged
fluorescence decay lifetimes, $\tau_{D(A),f}$. These polynomials take the form: $\tau_{D(A),xj} = \sum_{i=0}^{4} p_i (\tau_{D(A),fj})^i$.

Polynomial coefficients are determined via simulations of state widths in the Margarita software
package. Static FRET-lines describe the expected relationship between $\tau_{D(A),f}$ and FRET efficiency
for non-dynamic populations as $\tau_{D(A),f}$ is varied from 0 to the donor-only lifetime. Values for $\tau_{D(0)}$ are
listed in *Appendix 1—table 3*. Details of variants can be found in *Figure 1* and *Table 1*. The static
FRET-lines take the form: $E = 1 - (\sum_{i=0}^{4} \frac{p_i (\tau_{D(A),f})^i}{\tau_{D(0)}})$.

| FL Variant | $p_0$ | $p_1$ | $p_2$ | $p_3$ | $p_4$ |
|---|---|---|---|---|---|
| P2-G6 | –0.020 | 0.362 | 0.465 | –0.109 | 0.008 |
| P3-G4 | –0.020 | 0.368 | 0.442 | –0.099 | 0.007 |
| P2-G1 | –0.020 | 0.362 | 0.466 | –0.109 | 0.008 |
| P3-G5 | –0.021 | 0.372 | 0.430 | –0.094 | 0.006 |
| P1-G3 | –0.019 | 0.361 | 0.471 | –0.111 | 0.008 |
| P1-G4 | –0.021 | 0.369 | 0.438 | –0.098 | 0.007 |
| P1-G2 | –0.020 | 0.369 | 0.441 | –0.099 | 0.007 |
| P3-S2 | –0.022 | 0.378 | 0.407 | –0.086 | 0.005 |
| P2-S2 | –0.018 | 0.353 | 0.500 | –0.124 | 0.009 |
| S2-G6 | –0.019 | 0.358 | 0.481 | –0.115 | 0.008 |
| S3-G1 | –0.019 | 0.361 | 0.471 | –0.111 | 0.008 |
| S1-G1 | –0.020 | 0.364 | 0.459 | –0.106 | 0.008 |

| PSG Variant | $p_0$ | $p_1$ | $p_2$ | $p_3$ | $p_4$ |
|---|---|---|---|---|---|
| P2-G1 | –0.020 | 0.362 | 0.465 | –0.109 | 0.008 |
| P3-G5 | –0.020 | 0.362 | 0.466 | –0.109 | 0.008 |
| P1-G3 | –0.020 | 0.367 | 0.448 | –0.102 | 0.007 |
| P1-G4 | –0.019 | 0.357 | 0.486 | –0.118 | 0.009 |
| P1-G2 | –0.020 | 0.369 | 0.441 | –0.099 | 0.007 |
| P3-S2 | –0.021 | 0.372 | 0.427 | –0.093 | 0.006 |

**Appendix 1—table 6.** Parameters for calculation of dynamic FRET-lines in MFD plots.
The dynamic FRET-lines are calculated using these parameters and shown for visual reference in
MFD plots in *Figure 3* and *Figure 4—figure supplement 2*. Two-State dynamic FRET-lines describe
the path in FRET efficiency vs $\tau_{D(A),f}$ plots on which populations exhibiting fractional mixing between

two limiting, Gaussian-distributed states would fall. Dynamic lines are corrected like static lines for linker dynamics. Lifetimes for limiting states are taken from seTCSPC analysis (*Figure 3A* and *Appendix 1—table 3*). Details of variants can be found in *Figure 1* and *Table 1*. Dynamic FRET-lines take the form: $E = 1 - \frac{(\tau_{D(A),\,fI})(\tau_{D(A),\,fII})}{\tau_{D(0)}(\tau_{D(A).fI} + \tau_{D(A).fII} - \sum_{i=0}^{3} p_i(\tau_{D(A).f})^i)}$.

| Variant | $p_0$ | $p_1$ | $p_2$ | $p_3$ |
|---------|-------|-------|-------|-------|
| P2-G6 | −2.499 | 1.894 | 0.000 | 0.000 |
| P3-G4 | −2.134 | 1.768 | 0.000 | 0.000 |
| P2-G1 | −2.608 | 1.767 | 0.000 | 0.000 |
| P3-G5 | −2.653 | 2.023 | 0.000 | 0.000 |
| P1-G3 | −1.355 | 1.400 | 0.000 | 0.000 |
| P1-G4 | −1.482 | 1.414 | 0.000 | 0.000 |
| P1-G2 | −4.946 | 2.435 | 0.000 | 0.000 |
| P3-S2 | −3.337 | 2.453 | 0.000 | 0.000 |
| P2-S2 | −1.984 | 1.907 | 0.000 | 0.000 |
| S2-G6 | −2.118 | 2.049 | 0.000 | 0.000 |
| S3-G1 | −1.675 | 1.560 | 0.000 | 0.000 |
| S1-G1 | −1.763 | 1.586 | 0.000 | 0.000 |

| Variant | $p_0$ | $p_1$ | $p_2$ | $p_3$ |
|---------|-------|-------|-------|-------|
| P2-G1 | −1.955 | 1.641 | 0.000 | 0.000 |
| P3-G5 | −2.146 | 2.003 | 0.000 | 0.000 |
| P1-G3 | −1.713 | 1.479 | 0.000 | 0.000 |
| P1-G4 | −2.014 | 1.597 | 0.000 | 0.000 |
| P1-G2 | −3.712 | 2.179 | 0.000 | 0.000 |
| P3-S2 | −2.247 | 1.844 | 0.000 | 0.000 |

# Appendix 2

**Appendix 2—table 1.** Reaction rates and population fractions from Photon Distribution Analysis (PDA).

Fit parameters from PDA. Fits were performed with 2 static and 1 dynamic population, corresponding to distances from seTCSPC Gaussian-distributed states for static populations and exchange between them for the dynamic population. Details of variants can be found in *Figure 1* and *Table 1*.

| FL Variant | $k_{AB}$ (ms$^{-1}$) | $k_{BA}$ (ms$^{-1}$) | $k_R$ (ms$^{-1}$) | $T_R$ (ms) | Static Fraction A (%) | Static Fraction B (%) | Kinetic Fraction (%) |
|---|---|---|---|---|---|---|---|
| P2-G6 | 4.10 | 3.15 | 7.25 | 0.14 | 0.4 | 4.3 | 95.3 |
| P3-G4 | 3.08 | 2.99 | 6.07 | 0.17 | 9.8 | 25.2 | 65.0 |
| P2-G1 | 4.76 | 3.56 | 8.32 | 0.12 | 1.4 | 8.1 | 90.5 |
| P3-G5 | 1.97 | 5.76 | 7.73 | 0.13 | 19.4 | 1.3 | 83.1 |
| P1-G3 | 4.53 | 7.00 | 11.53 | 0.09 | 25.0 | 8.0 | 67.1 |
| P1-G4 | 3.25 | 4.75 | 8.00 | 0.13 | 0.0 | 25.1 | 74.9 |
| P1-G2 | 0.72 | 8.11 | 8.83 | 0.11 | 27.7 | 0.4 | 72.0 |
| P3-S2 | 5.54 | 3.14 | 8.68 | 0.12 | 6.5 | 3.2 | 90.3 |
| P2-S2 | 2.00 | 7.03 | 9.03 | 0.11 | 27.8 | 1.0 | 71.2 |
| S2-G6 | 7.32 | 5.01 | 12.33 | 0.08 | 2.9 | 0.0 | 97.1 |
| S3-G1 | 5.19 | 6.76 | 11.95 | 0.08 | 11.2 | 2.2 | 86.6 |
| S1-G1 | 3.52 | 5.84 | 9.36 | 0.11 | 1.7 | 14.2 | 84.1 |

| PSG Variant | $k_{AB}$ (ms$^{-1}$) | $k_{BA}$ (ms$^{-1}$) | $k_R$ (ms$^{-1}$) | $T_R$ (ms) | Static Fraction A (%) | Static Fraction B (%) | Kinetic Fraction (%) |
|---|---|---|---|---|---|---|---|
| P2-G1 | 5.09 | 10.80 | 15.89 | 0.06 | 35.4 | 19.2 | 45.4 |
| P3-G5 | 4.10 | 7.65 | 11.75 | 0.09 | 23.7 | 26.3 | 50.0 |
| P1-G3 | 5.28 | 15.90 | 21.18 | 0.05 | 35.0 | 3.3 | 61.7 |
| P1-G4 | 3.27 | 16.20 | 19.47 | 0.05 | 26.4 | 2.9 | 70.7 |
| P1-G2 | 2.91 | 28.80 | 31.71 | 0.03 | 35.8 | 0.1 | 64.1 |
| P3-S2 | 7.32 | 4.23 | 11.55 | 0.09 | 11.0 | 13.6 | 75.4 |

# Appendix 3

**Appendix 3—table 1.** Fit parameters from global analysis of filtered Fluorescence Correlation Spectroscopy (fFCS).

Cross-correlation decay times were fixed for all samples, and their amplitudes were set as global parameters when fitting the cross-correlation curves. The average and log-space average, diffusion time ($t_{diff.}$) and geometric factor s are also global fit parameters as defined in Materials and Methods. Details of variants can be found in *Figure 1* and *Table 1*. Decay timescales correspond to $t_R$ and amplitudes to $CC_i$ in Materials and Methods.

| FL Variant | 0.01ms Amplitude | 0.10ms Amplitude | 1.00ms Amplitude | Average | $t_{diff.}$ (ms) | S |
|---|---|---|---|---|---|---|
| P2-G6 | 0.02 | 0.37 | 0.61 | 0.65 | 8.64 | 4.70 |
| P3-G4 | 0.53 | 0.31 | 0.16 | 0.20 | 10.19 | 4.70 |
| P2-G1 | 0.54 | 0.01 | 0.45 | 0.46 | 2.27 | 4.42 |
| P3-G5 | 0.00 | 0.50 | 0.50 | 0.55 | 4.47 | 4.42 |
| P1-G3 | 0.23 | 0.17 | 0.60 | 0.62 | 0.91 | 4.42 |
| P1-G4 | 0.53 | 0.00 | 0.47 | 0.47 | 1.08 | 4.42 |
| P1-G2 | 0.47 | 0.13 | 0.39 | 0.41 | 2.05 | 4.42 |
| P3-S2 | 0.27 | 0.10 | 0.62 | 0.64 | 0.91 | 4.42 |
| P2-S2 | 0.03 | 0.41 | 0.56 | 0.60 | 7.03 | 4.70 |
| S2-G6 | 0.01 | 0.28 | 0.71 | 0.74 | 9.74 | 4.70 |
| S3-G1 | 0.63 | 0.00 | 0.37 | 0.37 | 1.21 | 4.70 |
| S1-G1 | 0.71 | 0.03 | 0.26 | 0.27 | 2.10 | 4.70 |

| PSG Variant | 0.01ms Amplitude | 0.10ms Amplitude | 1.00ms Amplitude | Average | $t_{diff.}$ (ms) | S |
|---|---|---|---|---|---|---|
| P2-G1 | 0.39 | 0.03 | 0.58 | 0.59 | 1.04 | 4.42 |
| P3-G5 | 0.29 | 0.18 | 0.53 | 0.55 | 0.79 | 4.42 |
| P1-G3 | 0.51 | 0.10 | 0.39 | 0.41 | 1.54 | 4.42 |
| P1-G4 | 0.58 | 0.00 | 0.42 | 0.43 | 0.80 | 4.42 |
| P1-G2 | 0.39 | 0.46 | 0.15 | 0.20 | 3.12 | 4.42 |
| P3-S2 | 0.01 | 0.31 | 0.68 | 0.72 | 0.96 | 4.42 |

**Appendix 3—table 2.** Variant-specific fit parameters for analysis of filtered Fluorescence Correlation Spectroscopy (fFCS).

The baseline/no-correlation parameter B, average number of bright molecules in the confocal volume N, and long-timescale decay amplitude ($A_{Tl}$ for $t_L$ = 5.00 ms, accounts for long-timescale photophysical effects). Parameters correspond to low-FRET to high-FRET cross-correlation (LH), high-FRET to low-FRET cross-correlation (HL), low-FRET autocorrelation (LL), or high-FRET autocorrelation (HH). Details of variants can be found in *Figure 1* and *Table 1*. Parameter definitions can be found in Materials and Methods.

| FL Variant | $b_{LH}$ | $N_{B,LH}$ | $A_{Tl,LH}$ | $b_{HL}$ | $N_{B,HL}$ | $A_{Tl,HL}$ | $b_{LL}$ | $N_{B,LL}$ | $A_{Tl,LL}$ | $b_{HH}$ | $N_{B,HH}$ | $A_{Tl,HH}$ |
|---|---|---|---|---|---|---|---|---|---|---|---|---|
| P2-G6 | 1.00 | 0.03 | 0.00 | 1.00 | 0.04 | 0.00 | 1.00 | 0.03 | 0.00 | 1.00 | 0.01 | 0.00 |
| P3-G4 | 1.02 | 37.23 | 0.59 | 1.02 | 4.85 | 0.01 | 1.02 | 0.06 | 0.78 | 1.02 | 0.03 | 0.89 |
| P2-G1 | 1.14 | 0.57 | 0.00 | 1.14 | 0.34 | 0.41 | 1.24 | 0.05 | 0.19 | 1.09 | 0.15 | 0.03 |
| P3-G5 | 1.03 | 1.10 | 0.12 | 1.03 | 0.96 | 0.23 | 1.01 | 0.06 | 0.27 | 1.00 | 0.21 | 0.20 |
| P1-G3 | 1.00 | 0.12 | 0.02 | 1.00 | 0.11 | 0.00 | 1.00 | 0.05 | 0.00 | 1.00 | 0.02 | 0.00 |

*Appendix 3—table 2 Continued on next page*

*Appendix 3—table 2 Continued*

| FL Variant | $b_{LH}$ | $N_{B,LH}$ | $A_{TI,LH}$ | $b_{HL}$ | $N_{B,HL}$ | $A_{TI,HL}$ | $b_{LL}$ | $N_{B,LL}$ | $A_{TI,LL}$ | $b_{HH}$ | $N_{B,HH}$ | $A_{TI,HH}$ |
|---|---|---|---|---|---|---|---|---|---|---|---|---|
| P1-G4 | 1.24 | 0.20 | 0.06 | 1.24 | 0.26 | 0.00 | 1.32 | 0.06 | 0.00 | 1.20 | 0.01 | 0.00 |
| P1-G2 | 1.05 | 0.16 | 0.12 | 1.05 | 0.18 | 0.00 | 1.28 | 0.02 | 0.12 | 1.05 | 0.03 | 0.00 |
| P3-S2 | 1.35 | 0.14 | 0.06 | 1.35 | 0.10 | 0.29 | 1.45 | 0.03 | 0.00 | 1.50 | 0.05 | 0.00 |
| P2-S2 | 1.00 | 0.32 | 0.01 | 1.00 | 0.78 | 0.00 | 1.00 | 0.02 | 0.00 | 1.00 | 0.16 | 0.00 |
| S2-G6 | 1.00 | 0.82 | 0.00 | 1.00 | 0.95 | 0.00 | 1.00 | 0.43 | 0.00 | 1.00 | 0.51 | 0.00 |
| S3-G1 | 1.00 | 1.49 | 0.00 | 1.00 | 0.90 | 0.00 | 1.00 | 0.06 | 0.00 | 1.00 | 0.05 | 0.00 |
| S1-G1 | 1.00 | 1.99 | 0.00 | 1.00 | 1.20 | 0.21 | 1.00 | 0.05 | 0.04 | 1.00 | 0.05 | 0.04 |

| PSG Variant | $B_{LH}$ | $N_{LH}$ | $A_{TI,LH}$ | $B_{HL}$ | $N_{HL}$ | $A_{TI,HL}$ | $B_{LL}$ | $N_{LL}$ | $A_{TI,LL}$ | $B_{HH}$ | $N_{HH}$ | $A_{TI,HH}$ |
|---|---|---|---|---|---|---|---|---|---|---|---|---|
| P2-G1 | 1.05 | 0.09 | 0.57 | 1.05 | 0.12 | 0.00 | 1.11 | 0.01 | 0.00 | 1.03 | 0.04 | 0.00 |
| P3-G5 | 1.25 | 0.19 | 0.31 | 1.25 | 0.16 | 0.37 | 1.85 | 0.01 | 0.01 | 1.15 | 0.04 | 0.00 |
| P1-G3 | 1.18 | 0.43 | 0.00 | 1.18 | 0.48 | 0.00 | 1.20 | 0.08 | 0.00 | 1.17 | 0.02 | 0.00 |
| P1-G4 | 1.10 | 0.23 | 0.00 | 1.10 | 0.24 | 0.00 | 1.10 | 0.05 | 0.00 | 1.10 | 0.01 | 0.00 |
| P1-G2 | 1.17 | 0.08 | 0.04 | 1.17 | 0.07 | 0.19 | 1.34 | 0.01 | 0.23 | 1.10 | 0.03 | 0.05 |
| P3-S2 | 1.16 | 0.20 | 0.00 | 1.16 | 0.16 | 0.24 | 1.40 | 0.01 | 0.00 | 1.07 | 0.07 | 0.00 |

# Appendix 4

**Appendix 4—table 1.** Attachment atom indices for accessible volume simulations.
Fluorophores were simulated in the FPS software package using a three-sphere model (Radii R1, R2, R3), with flexible linker dimensions given by LL (length) and WL (width). Docking simulations were performed with Alexa-488 C5-maleimide attached to PDZ3 sites and Alexa-647 C2-maleimide attached to SH3-GuK. Parameters for accessible volume simulations used in rigid body docking and simulations were as follows: For Alexa-488 C5-maleimide, radii values of 5.0 Å, 4.5 Å, and 1.5 Å were used with linker length of 20.5 Å and width 4.5 Å. For Alexa-674 C2-maleimide, radii values of 7.15 Å, 4.5 Å, and 1.5 Å were used with linker length of 21.0 Å and width 4.5 Å. Details of variants can be found in *Figure 1* and *Table 1*.

| Labeling Site | Atom Index |
|---|---|
| **PDZ3** | |
| P1 | 14 |
| P2 | 554 |
| P3 | 778 |
| **SH3-GuK** | |
| S1 | 519 |
| S2 | 664 |
| S3 | 870 |
| G1 | 1680 |
| G2 | 1830 |
| G3 | 1948 |
| G4 | 1972 |
| G5 | 2157 |
| G6 | 2477 |

**Appendix 4—table 2.** Distance bounds from FRET network robustness analysis.
Mean distances from re-analysis of sub-sampled seTCSPC data are reported for the two limiting states from each variant (*Figure 8—figure supplement 1*). These distances, along with AV simulations, were used for screening all snapshot structures from DMD simulations to estimate FRET-distances for each snapshot structure. Details of variants can be found in *Figure 1* and *Table 1*. For inclusion in the basin ensembles from screening, simulated structures were rejected if the percent error of each distance was greater than the average σ%error from the corresponding basin (7.3% and 8.7% for limiting states A and B, respectively). Description of FRET Network Robustness Analysis is found in Materials and methods.

| Sample | ± $\sigma_{\%error}$ A (Å) | ± $\sigma_{\%error}$ B (Å) |
|---|---|---|
| P2-G6 | 68.1±5.0 | 37.7±3.3 |
| P3-G4 | 66.3±4.8 | 35.6±3.1 |
| P2-G1 | 81.6±6.0 | 34.5±3.0 |
| P3-G5 | 64.3±4.7 | 41.0±3.6 |
| P1-G3 | 85.8±6.3 | 41.4±3.6 |
| P1-G4 | 75.4±5.5 | 34.1±3.0 |
| P1-G2 | 77.3±5.6 | 27.2±2.4 |
| P3-S2 | 28.3±2.1 | 52.0±4.5 |

*Appendix 4—table 2 Continued on next page*

*Appendix 4—table 2 Continued*

| Sample | ± σ$_{\%error}$ A (Å) | ± σ$_{\%error}$ B (Å) |
|---|---|---|
| P2-S2 | 36.9±2.7 | 59.1±5.1 |
| S2-G6 | 33.4±2.4 | 53.5±4.7 |
| S3-G1 | 38.8±2.8 | 73.2±6.4 |
| S1-G1 | 65.9±4.8 | 33.7±2.9 |

