## [Editor Report]

This paper presents data of fundamental importance and is methodologically exceptional. It is of broad interest to investigators studying the function and regulation of protein scaffolds, dynamic protein structure, and the regulation of the postsynaptic density at excitatory synapses. The authors develop an integrated approach using fluorescence-based biochemical methods, disulfide mapping, and discrete molecular dynamic simulations to study the dynamic supertertiary conformation of the synaptic scaffold protein PSD-95. The overall research strategy serves as a textbook example to the field.

---

## [Decision Letter]

**Decision letter after peer review:**

Thank you for submitting your article "Fuzzy Supertertiary Interactions within PSD-95 Enable Ligand Binding" for consideration by *eLife*. Your article has been reviewed by 3 peer reviewers, including Mary B Kennedy as Reviewing Editor and Reviewer #1, and the evaluation has been overseen by Volker Dötsch as the Senior Editor. The following individuals involved in the review of your submission have agreed to reveal their identity: Jelle Hendrix (Reviewer #2); Andrew Lee (Reviewer #3).

Essential revisions:

1. Although this study is important and even ground-breaking, it is very difficult to read as presently written. The authors need to take several steps to improve the clarity of the presentation, in particular for the broad audience of *eLife*.

The authors need to reduce the use of jargon throughout the Results section. In the Introduction, it would be useful to include a summary of the experimental trajectory that lists the major methods used and their acronym abbreviations so that readers can refer back to a single place to decode the acronyms as they read the results. For example, after your sentence, "Here we applied this approach to the PSG supramodule", add the following sentences: "We made multiparameter fluorescence measurements (MFM) in which single-molecular Fluorescent Resonance Energy Transfer (FRET) data were acquired by sub-ensemble Time-Correlated Single Photon Counting (seTCSPC). We used Total Internal Reflectance Fluorescence microscopy (smTIRF) to compare fluorescence measurements from truncated PSG supramodules to those from full-length PSD-95. Measurements were analyzed by Photon Distribution Analysis (PDA) to confirm the existence of two limiting energy states. To resolve fast dynamics, we performed filtered Fluorescence Correlation Spectroscopy (fFCS). To fully map and refine the conformational energy landscape of the supramodule, we performed a replica exchange Discrete Molecular Dynamics (DMD). Predicted conformations were verified by disulfide (DS) mapping after the introduction of pairs of cysteine mutations. Finally, we used the FRET restraints to simulate the accessible volume (AV) for dye pairs at each labeling site."

2. The authors need to carefully explain the utility of each of the experimental, statistical, and simulation methods that they use at each point. For example, they should include in the introduction a summary of the flow of the analysis, perhaps a shortened version of the summary in the first part of the public review.

a. The discussion of almost every figure should start with a process-like overview of the different steps that are performed to arrive at the final result; when appropriate, including a diagram of the analysis as part A of the figure.

b. Many of the supplemental figures (even if only exemplary subpanels thereof), should be moved to the main text. Supplemental material should not contain essential elements that are needed to understand the main text. In view of this, at a minimum, Figure 1 Suppl 1, parts of Figure 3 Supplements 1 and 2, and parts of Figure 6 Supplements 1 and 2 should be moved to the main text. *eLife* does not have limits on the number of main figures needed to justify the conclusions of a study.

3. The authors should explain clearly why they used two different methods to integrate fluorescence data with modeling data (rigid body modeling and DMD snapshot screening with FNR analysis). What is meant exactly by the statement, "For state A, the best-fit models showed some divergence with PDZ3 near the HOOK insertion but also extended without interdomain contacts (Figure 6A)." What divergence are they referring to?

4. In order to interpret the disulfide mapping experiment correctly, the authors need to show data from negative controls testing cysteine pairs that are predicted NOT to interact.

5. The authors need to explain how the approach taken in this paper compares to their previous simulated annealing approach of mapping PDZ3 using FRET data in their McCann et al., 2012 paper. That study resulted in a model in which PDZ3 binds to a completely different interface, which is not mentioned in this manuscript.

6. The authors need to discuss whether some of the FRET variants may be undergoing increased domain-swap dimerization compared to the wild type. Might this occurence have influenced their data?

7. p. 17, line 32: A proper reference should be given here, probably after "10-phenanthroline".

8. p. 13, line 4: "residues 303-274" is a typo and should be fixed.

9. Small typo in Figure 5F legend: (%/min) => (/min)?

10. Supp file 3A: caption described 'average' and log-average, yet the table only describes average. Please also change 'Average' to the T_R to be consistent with PDA, and describe clearly how it is calculated.

---

## [Author Response]

Essential revisions:1. Although this study is important and even ground-breaking, it is very difficult to read as presently written. The authors need to take several steps to improve the clarity of the presentation, in particular for the broad audience of eLife.

We thank the reviewers for their suggestions as to how we can best improve the clarity of the manuscript given the broad array of methods and analyses, which each carry their own acronyms and jargon. We have added a couple paragraphs to the Introduction that explains each method and defines the acronyms, which can serve as a reference point. We have also included a new workflow diagram as Figure 2, which clarifies how the different methods are connected to the analyses in use. We have also made changes to the manuscript to use more common language in place of technical jargon and reduce the number acronyms used in the text.

2. The authors need to carefully explain the utility of each of the experimental, statistical, and simulation methods that they use at each point. For example, they should include in the introduction a summary of the flow of the analysis, perhaps a shortened version of the summary in the first part of the public review.a. The discussion of almost every figure should start with a process-like overview of the different steps that are performed to arrive at the final result; when appropriate, including a diagram of the analysis as part A of the figure.b. Many of the supplemental figures (even if only exemplary subpanels thereof), should be moved to the main text. Supplemental material should not contain essential elements that are needed to understand the main text. In view of this, at a minimum, Figure 1 Suppl 1, parts of Figure 3 Supplements 1 and 2, and parts of Figure 6 Supplements 1 and 2 should be moved to the main text. eLife does not have limits on the number of main figures needed to justify the conclusions of a study.

We appreciate the opportunity to move more data and analysis to the main text. Specifically, we have increase the number of main text figures from 7 to 10 and moved one of the supplemental tables to the main text. We thank the reviewers for their suggestions as to which panels were key to understanding the manuscript and have moved all the suggested panels to the main text. This includes the former Figure 1—figure supplements 1, 2 and 3; Figure 3—figure supplement 2; along with Figure 6—figure supplement 2.

3. The authors should explain clearly why they used two different methods to integrate fluorescence data with modeling data (rigid body modeling and DMD snapshot screening with FNR analysis). What is meant exactly by the statement, "For state A, the best-fit models showed some divergence with PDZ3 near the HOOK insertion but also extended without interdomain contacts (Figure 6A)." What divergence are they referring to?

We thank the reviewers for this suggestion to improve the clarity regarding our use of two different methods to generate models based on the FRET data. We see the two methods as complementary with rigid body docking representing a standard front line method for structural modeling based on FRET data while the DMD snapshot screening allowed us to provide more details about the structural ensemble for each of the predominant states and also to test the robustness and thereby resolution of the FRET model. Rigid body docking relied only on distances obtained from fitting experimental data, which is often the case in FRET studies. The scoring functions in rigid body docking identify a single best structure (or family of structures) for a given set of distance restraints. In this work, all 12 variants were globally fit to assign the experimental distances to their corresponding states. This allowed us to identify a small number of top scoring models for each set of distance restraints.

In this study, we had an independent dataset of structural models from DMD simulations, which is not always the case in FRET studies. The DMD ensemble, along with FRET Network Robustness Analysis, allowed us to revisit the fluorescence data to validate the experimental models from rigid body docking and also to establish the responsiveness of individual variants to the underlying conformational distribution. Additionally, because the DMD generates such a well-sampled energy landscape compared to rigid body docking, selecting models from the DMD ensemble provided a more complete picture of the basin fuzziness than is possible from handful of structures.

By using these two methods along with disulfide mapping, we provide validation that rigid body docking can accurately resolve dynamic structures, given sufficient time resolution relative to the conformational exchange. We also show that well equilibrated DMD simulations can be used for the purposes of FRET modeling and resolve finer details of the basin fuzziness under experimental conditions in addition to providing estimations of the model uncertainty.

Regarding the quoted text, we erroneously referred to the differences between modeling approaches as “divergence”. We meant that in addition to the presence of similar models from rigid body docking and DMD screening, which position PDZ3 near HOOK, the DMD screening identified models with PDZ3 farther away from the HOOK, which did not occur amongst the top ranked structures from rigid body docking. We have corrected this in the manuscript.

4. In order to interpret the disulfide mapping experiment correctly, the authors need to show data from negative controls testing cysteine pairs that are predicted NOT to interact.

We agree that controls are a critical part of the disulfide mapping experiments and thank the reviewers for this suggestion. As a negative control, we selected a cysteine pair that showed low FRET in our 2012 PNAS paper (Q374C-K591C), which was not included in this work. This cysteine pair should not be involved in contact interfaces identified from simulations or modeling. This cysteine pair showed no evidence of intramolecular disulfide formation. In the manuscript, we have added discussion of this negative control and provide an additional figure (Figure 7—figure supplement 2) to document that this variant does not form disulfides.

5. The authors need to explain how the approach taken in this paper compares to their previous simulated annealing approach of mapping PDZ3 using FRET data in their McCann et al., 2012 paper. That study resulted in a model in which PDZ3 binds to a completely different interface, which is not mentioned in this manuscript.

We apologize for this oversight and thank the reviewers for bringing it to our attention. The original modeling used rigid body docking, which was very similar to the approach used herein. The main problem with our earlier smTIRF model was that the experimental data was collected with insufficient time resolution to resolve the two conformational basins. Each FRET restraint reported the time averaged distance for that pair, but as our current work shows, these variants have independent dynamic signatures that depend on the specific structural interconversion pathways of the domains and the experimental sensitivity of each FRET pair. We have included this commentary on our previous model in the revised discussion on page 9.

6. The authors need to discuss whether some of the FRET variants may be undergoing increased domain-swap dimerization compared to the wild type. Might this occurence have influenced their data?

We are very interested in the prospect of domain swapping as has been suggested previously. However, we have not seen evidence for this at the concentrations used here. As reported in our 2012 PNAS paper, both full-length PSD-95 and the PSG fragment are monodisperse as judged by size exclusion chromatography, which suggests that lack of stably populated oligomeric states under these conditions at 10^-5^ molar concentrations. The PSG fragment runs very true to its calculated formula weight while the full-length protein does migrate faster than expected based on formula weight but not high enough to be a multimer.

The DS mapping experiments did reveal some higher molecular weight species. However, these higher order species never accounted for more than 5% of the total input. Thus, any homomeric, intramolecular interaction is transient and not well occupied under the buffer conditions and concentrations used in these studies. Our size exclusion and disulfide mapping experiments are carried out at protein concentrations that are orders of magnitude higher than used for single molecule imaging. Thus, dimerization is unlikely at the single-molecule concentrations used for the present FRET experiments. If dimerization were to occur, we would expect the appearance of additional static subpopulations in the MFD histograms. If dimerization were significant, we would also expect the appearance of an additional diffusion term in fluorescence correlation curves, which was not the case in these experiments.